# Nonparametric Density Estimation and Convergence of GANs under Besov IPM Losses

**Ananya Uppal**
Department of Mathematical Sciences
Carnegie Mellon University
auppal@andrew.cmu.edu

**Shashank Singh**[*]  **Barnabás Póczos**
Machine Learning Department
Carnegie Mellon University
{sss1,bapoczos}@cs.cmu.edu

## Abstract

We study the problem of estimating a nonparametric probability density under a large family of losses called Besov IPMs, which include, for example, $L^p$ distances, total variation distance, and generalizations of both Wasserstein and Kolmogorov-Smirnov distances. For a wide variety of settings, we provide both lower and upper bounds, identifying precisely how the choice of loss function and assumptions on the data interact to determine the minimax optimal convergence rate. We also show that linear distribution estimates, such as the empirical distribution or kernel density estimator, often fail to converge at the optimal rate. Our bounds generalize, unify, or improve several recent and classical results. Moreover, IPMs can be used to formalize a statistical model of generative adversarial networks (GANs). Thus, we show how our results imply bounds on the statistical error of a GAN, showing, for example, that GANs can strictly outperform the best linear estimator.

## 1 Introduction

This paper studies the problem of estimating a nonparametric probability density, using an integral probability metric as a loss. That is, given a sample space $\mathcal{X} \subseteq \mathbb{R}^D$, suppose we observe $n$ IID samples $X_1, ..., X_n \overset{IID}{\sim} p$ from a probability density $p$ over $\mathcal{X}$ that is unknown but assumed to lie in a regularity class $\mathcal{P}$. We seek an estimator $\widehat{p} : \mathcal{X}^n \to \mathcal{P}$ of $p$, with the goal of minimizing a loss

$$d_{\mathcal{F}}(p, \widehat{p}(X_1, ..., X_n)) := \sup_{f \in \mathcal{F}} \left| \mathbb{E}_{X \sim p} [f(X)] - \mathbb{E}_{X \sim \widehat{p}(X_1, ..., X_n)} [f(X)] \right|, \qquad (*)$$

where $\mathcal{F}$, called the *discriminator class*, is some class of bounded, measurable functions on $\mathcal{X}$.

Metrics of the form $(*)$ are called *integral probability metrics* (IPMs), or $\mathcal{F}$-IPMs[2], and can capture a wide variety of metrics on probability distributions by choosing $\mathcal{F}$ appropriately [38]. This paper studies the case where both $\mathcal{F}$ and $\mathcal{P}$ belong to the family of Besov spaces, a large family of nonparametric smoothness spaces that include, as examples, $L^p$, Lipschitz/Hölder, and Hilbert-Sobolev spaces. The resulting IPMs include, as examples, $L^p$, total variation, Kolmogorov-Smirnov, and Wasserstein distances. We have two main motivations for studying this problem:

1. This problem unifies nonparametric density estimation with the central problem of empirical process theory, namely bounding quantities of the form $d_{\mathcal{F}}(P, \widehat{P})$ when $\widehat{P}$ is the empirical distribution $P_n = \frac{1}{n} \sum_{i=1}^{n} \delta_{X_i}$ of the data [42]. Whereas empirical process theory typically avoids restricting $\mathcal{P}$ and fixes the estimator $\widehat{P} = P_n$, focusing on the discriminator class $\mathcal{F}$, nonparametric density estimation typically fixes the loss to be an $L^p$ distance, and seeks a good estimator $\widehat{P}$ for a given

---

[*]Now at Google.

[2]While the name IPM seems most widely used [38, 48, 6, 58], many other names have been used for these quantities, including *adversarial loss* [46, 12], *MMD* [16], and $\mathcal{F}$-*distance* or *neural net distance* [5].

distribution class $\mathcal{P}$. In contrast, we study how constraints on $\mathcal{F}$ and $\mathcal{P}$ *jointly* determine convergence rates of a number of estimates $\widehat{P}$ of $P$. In particular, since Besov spaces comprise perhaps the largest commonly-studied family of nonparametric function spaces, this perspective allows us to unify, generalize, and extend several classical and recent results in distribution estimation (see Section 3).

2. This problem is a theoretical framework for analyzing generative adversarial networks (GANs). Specifically, given a GAN whose discriminator and generator networks encode functions in $\mathcal{F}$ and $\mathcal{P}$, respectively, recent work [31, 27, 28, 46] showed that a GAN can be seen as a distribution estimate[3]

$$\widehat{P} = \operatorname*{argmin}_{Q \in \mathcal{P}} \sup_{f \in \mathcal{F}} \left| \mathbb{E}_{X \sim Q} [f(X)] - \mathbb{E}_{X \sim \widetilde{P}_n} [f(X)] \right| = \operatorname*{argmin}_{Q \in \mathcal{P}} d_{\mathcal{F}} \left( Q, \widetilde{P}_n \right), \qquad (1)$$

i.e., an estimate which directly minimizes empirical IPM risk with respect to a (regularized) empirical distribution $\widetilde{P}_n$. While, in the original GAN model [20], $\widetilde{P}_n$ was the empirical distribution $P_n = \frac{1}{n} \sum_{i=1}^{n} \delta_{X_i}$ of the data, Liang [27] showed that, under smoothness assumptions on the population distribution, performance is improved by replacing $P_n$ with a regularized version $\widetilde{P}_n$, equivalent to the instance noise trick that has become standard in GAN training [47, 34]. We show, in particular, that, when $\widetilde{P}_n$ is a wavelet-thresholding estimate, a GAN based on sufficiently large fully-connected neural networks with ReLU activations learns Besov probability distributions at the optimal rate.

## 2 Set up and Notation

For non-negative real sequences $\{a_n\}_{n \in \mathbb{N}}$, $\{b_n\}_{n \in \mathbb{N}}$, $a_n \lesssim b_n$ indicates $\limsup_{n \to \infty} \frac{a_n}{b_n} < \infty$, and $a_n \asymp b_n$ indicates $a_n \lesssim b_n \lesssim a_n$. For $p \in [1, \infty]$, $p' := \frac{p}{p-1}$ denotes the Hölder conjugate of $p$ (with $1' = \infty$, $\infty' = 1$). $L^p(\mathbb{R}^D)$ (resp. $l^p$) denotes the set of functions $f$ (resp. sequences $a$) with $\|f\|_p := \left( \int |f(x)|^p \, dx \right)^{1/p} < \infty$ (resp. $\|a\|_{l^p} := \left( \sum_{n \in \mathbb{N}} |a_n|^p \right)^{1/p} < \infty$).

### 2.1 Multiresolution Approximation and Besov Spaces

We now provide some notation that is necessary to define the family of Besov spaces studied in this paper. Since the statements and formal justifications behind these definitions are a bit complex, some technical details are relegated to the Appendix, and several well-known examples from the rich class of resulting spaces are given in Section 3. The diversity of Besov spaces arises from the fact that, unlike the Hölder or Sobolev spaces that they generalize, Besov spaces model functions simultaneously across multiple spatial scales. In particular, they rely on the following notion:

**Definition 1.** A *multiresolution approximation (MRA)* of $L^2(\mathbb{R}^D)$ is an increasing sequence $\{V_j\}_{j \in \mathbb{Z}}$ of closed linear subspaces of $L^2(\mathbb{R}^D)$ with the following properties:

1. $\bigcap_{j=-\infty}^{\infty} V_j = \{0\}$, and the closure of $\bigcup_{j=-\infty}^{\infty} V_j = L^2(\mathbb{R}^D)$.
2. For $f \in L^2(\mathbb{R}^D), k \in \mathbb{Z}^D, j \in \mathbb{Z}, f(x) \in V_0 \Leftrightarrow f(x-k) \in V_0 \ \& \ f(x) \in V_j \Leftrightarrow f(2x) \in V_{j+1}$.
3. For some "father wavelet" $\phi \in V_0$, $\{\phi(x-k) : k \in \mathbb{Z}^D\}$ is an orthonormal basis of $V_0 \subset L^2(\mathbb{R}^D)$.

For intuition, consider the best-known MRA of $L^2(\mathbb{R})$, namely the Haar wavelet basis. Let $\phi(x) = 1_{\{[0,1)\}}$ be the Haar father wavelet, let $V_0 = \operatorname{Span}\{\phi(x-k) : k \in \mathbb{Z}\}$ be the span of translations of $\phi$ by an integer, and let $V_j$ defined recursively for all $j \in \mathbb{Z}$ by $V_j = \{f(2x) : f(x) \in V_{j-1}\}$ be the set of horizontal scalings of functions in $V_{j-1}$ by $1/2$. Then, $\{V_j\}_{j \in \mathbb{Z}}$ is an MRA of $L^2(\mathbb{R})$.

The importance of an MRA is that it generates an orthonormal basis of $L^2(\mathbb{R}^D)$, via the following:

**Lemma 2** ([35], Section 3.9). *Let $\{V_j\}_{j \in \mathbb{Z}}$ be an MRA of $L^2(\mathbb{R}^D)$ with father wavelet $\phi$. Then, for $E = \{0,1\}^D \setminus (0, \dots, 0)$, there exist "mother wavelets" $\{\psi_\epsilon\}_{\epsilon \in E}$ such that $\{2^{Dj/2} \psi_\epsilon(2^j x - k) : \epsilon \in E, k \in \mathbb{Z}^D\} \cup \{2^{Dj/2} \phi(2^j x - k) : k \in \mathbb{Z}^D\}$ is an orthonormal basis of $V_j \subseteq L^2(\mathbb{R}^D)$.*

Let $\Lambda_j = \{2^{-j}k + 2^{-j-1}\epsilon : k \in \mathbb{Z}^D, \epsilon \in E\} \subseteq \mathbb{R}^D$. Then $k, \epsilon$ are uniquely determined for any $\lambda \in \Lambda_j$. Thus, for all $\lambda \in \Lambda := \bigcup_{j \in \mathbb{Z}} \Lambda_j$, we can let $\psi_\lambda(x) = 2^{Dj/2} \psi_\epsilon(2^j x - k)$. Equipped with the orthonormal basis $\{\psi_\lambda : \lambda \in \Lambda\}$ of $L^2(\mathbb{R}^D)$, we are almost ready to define Besov spaces.

For technical reasons (see, e.g., [35, Section 3.9]), we need MRAs of smoother functions than Haar wavelets, which are called *r-regular*. Due to space constraints, $r$-regularity is defined precisely in

Appendix A; we note here that standard $r$-regular MRAs exist, such as the Daubechies wavelet [10]. We assume for the rest of the paper that the wavelets defined above are supported on $[-A, A]$.

**Definition 3** (Besov Space). Let $0 \leq \sigma < r$, and let $p, q \in [1, \infty]$. Given an $r$-regular MRA of $L^2(\mathbb{R}^D)$ with father and mother wavelets $\phi, \psi$ respectively, the *Besov space* $B_{p,q}^\sigma(\mathbb{R}^D)$ is defined as the set of functions $f : \mathbb{R}^D \to \mathbb{R}$ such that, the wavelet coefficients

$$\alpha_k := \int_{\mathbb{R}^D} f(x)\phi(x-k)dx \ \text{ for } \ k \in \mathbb{Z}^D \quad \text{and} \quad \beta_\lambda := \int_{\mathbb{R}^D} f(x)\psi_\lambda(x)dx \ \text{ for } \ \lambda \in \Lambda,$$

$$\text{satisfy} \qquad \|f\|_{B_{p,q}^\sigma} := \|\{\alpha_k\}_{k \in \mathbb{Z}^D}\|_{l^p} + \left\| \left\{ 2^{j(\sigma + D(1/2 - 1/p))} \left\| \{\beta_\lambda\}_{\lambda \in \Lambda_j} \right\|_{l^p} \right\}_{j \in \mathbb{N}} \right\|_{l^q} < \infty$$

The quantity $\|f\|_{B_{p,q}^\sigma}$ is called the *Besov norm of $f$*, and, for any $L > 0$, we write $B_{p,q}^\sigma(L)$ to denote the closed Besov ball $B_{p,q}^\sigma(L) = \{f \in B_{p,q}^\sigma : \|f\|_{B_{p,q}^\sigma} \leq L\}$. When the constant $L$ is unimportant (e.g., for *rates* of convergence), $B_{p,q}^\sigma$ denotes a ball $B_{p,q}^\sigma(L)$ of finite but arbitrary radius $L$.

## 2.2 Formal Problem Statement

Having defined Besov spaces, we now formally state the statistical problem we study in this paper. Fix an $r$-regular MRA. We observe $n$ IID samples $X_1, ..., X_n \overset{IID}{\sim} p$ from an unknown probability density $p$ lying in a Besov ball $B_{p_g, q_g}^{\sigma_g}(L_g)$ with $\sigma_g < r$. We want to estimate $p$, measuring error with an IPM $d_{B_{p_d, q_d}^{\sigma_d}(L_d)}$. Specifically, for general $\sigma_d, \sigma_g, p_d, p_g, q_d, q_g$, we seek to bound minimax risk

$$M\left(B_{p_d, q_d}^{\sigma_d}, B_{p_g, q_g}^{\sigma_g}\right) := \inf_{\widehat{p}} \sup_{p \in B_{p_g, q_g}^{\sigma_g}} \underset{X_{1:n}}{\mathbb{E}} \left[ d_{B_{p_d, q_d}^{\sigma_d}}(p, \widehat{p}(X_1, \dots, X_n)) \right] \tag{2}$$

of estimating densities in $\mathcal{F}_g = B_{p_g, q_g}^{\sigma_g}$, where the infimum is taken over all estimators $\widehat{p}(X_1, \dots, X_n)$. In the rest of this paper, we suppress dependence of $\widehat{p}(X_1, ..., X_n)$ on $X_1, ..., X_n$, writing simply $\widehat{p}$.

# 3 Related Work

The current paper unifies, extends, or improves upon a number of recent and classical results in the nonparametric density estimation literature. Two areas of prior work are most relevant:

**Nonparametric estimation over inhomogeneous smoothness spaces** First is the classical study of estimation over inhomogeneous smoothness spaces under $L^p$ losses. Nemirovski [40] first noticed that, over classes of regression functions with inhomogeneous (i.e., spatially-varying) smoothness, many widely-used regression estimators, called "linear" estimators (defined precisely in Section 4.2), are provably unable to converge at the minimax optimal rate, in $L^2$ loss. Donoho et al. [13] identified a similar phenomenon for estimating probability densities in a Besov space $B_{p_g, q_g}^{\sigma_g}$ on $\mathbb{R}$ under $L^{p_d'}$ losses with $p_d' > p_g$, corresponding to the case $\sigma_d = 0, D = 1$ in our work. [13] also showed that the wavelet-thresholding estimator we consider in Section 4.1 *does* converge at the minimax optimal rate. We generalize these phenomena to many new loss functions; in many cases, linear estimators continue to be sub-optimal, whereas the wavelet-thresholding estimator continues to be optimal. We also show that sub-optimality of linear estimators is more pronounced in higher dimensions.

**Distribution estimation under IPMs** The second, more recent body of results [27, 46, 28] concerns nonparametric distribution estimation under IPM losses. Prior work focused on the case where $\mathcal{F}$ and $\mathcal{P}$ are both Sobolev ellipsoids, corresponding to the case $p_d = q_d = p_g = q_g = 2$ in our work. Notably, over these smaller spaces (of homogeneous smoothness), the linear estimators mentioned above are minimax rate-optimal. Perhaps the most important finding of these works is that the curse of dimensionality pervading classical nonparametric statistics is significantly diminished under weaker loss functions than $L^p$ losses (namely, many IPMs). For example, Singh et al. [46] showed that, when $\sigma_d > D/2$, one can estimate $P$ at the parametric rate $n^{-1/2}$ in the loss $d_{B_{2,2}^{\sigma_d}}$, without *any* regularity assumptions whatsoever on the probability distribution $P$. We generalize this to other losses $d_{B_{p_d, q_d}^{\sigma_d}}$.

These papers were motivated in part by a desire to understand theoretical properties of GANs, and, in particular, Liang [27] and Singh et al. [46] helped establish (1) as a valid statistical model of GANs. In particular, we note that Singh et al. [46] showed that the implicit generative modeling problem ("sampling") in terms of which GANs are usually framed, is equivalent, in terms of minimax

convergence rates, to nonparametric density estmation, justifying our focus on the latter problem in this paper. We show, in Section 4.3, that, given a sufficiently good optimization algorithm, GANs based on appropriately constructed deep neural networks can learn Besov densities at the minimax optimal rate. In this context, our results are among the first to suggest theoretically that GANs can outperform classical density estimators (namely, linear estimators mentioned above).

Liu et al. [31] provided general sufficient conditions for weak consistency of GANs in a generalization of the model (1). Since many IPMs, such as Wasserstein distances, metrize weak convergence of probability measures under mild additional assumptions Villani [52], this implies consistency under these IPMs. However, Liu et al. [31] did not study *rates* of convergence.

We end this section with a brief survey of known results for estimating distributions under specific Besov IPM losses, noting that our results (Equations (3) and (4) below) generalize all these rates:

1. $L^p$ **Distances:** If $\mathcal{F}_d = L^{p'} = B^0_{p',p'}$, then, for distributions $P, Q$ with densities $p, q \in L^p$, $d_{\mathcal{F}_d}(P, Q) = \|p - q\|_{L^p}$. These are the most well-studied losses in nonparametric statistics, especially for $p \in \{1, 2, \infty\}$ [41, 53, 51]. [13] studied the minimax rate of convergence of density estimation over Besov spaces under $L^p$ losses, obtaining minimax rates $n^{-\frac{\sigma_g}{2\sigma_g+D}} + n^{-\frac{\sigma_g+D(1-1/p_g-1/p_d)}{2\sigma_g+D(1-2/p_g)}}$ over general estimators, and $n^{-\frac{\sigma_g}{2\sigma_g+D}} + n^{-\frac{\sigma_g-D/p_g+D/p'_d}{2\sigma_g+D-2D/p_g+2D/p'_d}}$ when restricted to linear estimators.

2. **Wasserstein Distance:** If $\mathcal{F}_d = C^1(1) \asymp B^1_{\infty,\infty}$ is the space of 1-Lipschitz functions, then $d_{\mathcal{F}_d}$ is the 1-Wasserstein or Earth mover's distance (via the Kantorovich dual formulation [23, 52]). A long line of work has established convergence rates of the empirical distribution to the true distribution in spaces as general as unbounded metric spaces [54, 25, 45]). In the Euclidean setting, this is well understood [14, 2, 18], although, to the best of our knowledge, minimax lower bounds have been proven only recently [45]; this setting intersects with our work in the case $\sigma_d = 1, \sigma_g = 0$, $p_d = \infty$, matching our minimax rate of $n^{-1/D} + n^{-1/2}$. More general $p$-Wasserstein distances $W_p$ ($p \geq 1$) cannot be expressed exactly as IPMs, but, our results complement recent results of Weed and Berthet [55], who showed that, for densities $p$ and $q$ that are bounded above and below (i.e., $0 < m \leq p, q \leq M < \infty$), the bounds $M^{-1/p'} d_{B^1_{p',\infty}}(p, q) \leq W_p(p, q) \leq m^{-1/p'} d_{B^1_{p',1}}(p, q)$ hold; for such densities, our rates match theirs ($n^{-\frac{1+\sigma_g}{2\sigma_g+D}} + n^{-1/2}$) up to polylogarithmic factors. Weed and Berthet [55] showed that, without the lower-boundedness assumption ($m > 0$), minimax rates under $W_p$ are strictly slower (by a polynomial factor in $n$).

In machine learning applications, Arora et al. [5] recently used this rate to argue that, for data from a continuous distribution, Wasserstein GANs [4] cannot generalize at a rate faster than $n^{-1/D}$ (at least without additional regularization, as we use in Theorem 9). A variant in which $\mathcal{F}_d \subset C^1 \cap L^\infty$ is both uniformly bounded and 1-Lipschitz gives rise to the Dudley metric [15], which has also been suggested for use in GANs [1]. Finally, we note that the more general distances induced by $\mathcal{F}_d = B^{\sigma_d}_{\infty,\infty}$ have been useful for deriving central limit theorems [7, Section 4.8].

3. **Kolmogorov-Smirnov Distance:** If $\mathcal{F}_d = \text{BV} \asymp B^1_{1,\cdot}$ is the set of functions of bounded variation, then, in the 1-dimensional case, $d_{\mathcal{F}_d}$ is the well-known Kolmogorov-Smirnov metric [9], and so the famous Dvoretzky–Kiefer–Wolfowitz inequality [33] gives a parametric convergence rate of $n^{-1/2}$.

4. **Sobolev Distances:** If $\mathcal{F}_d = \mathcal{W}^{\sigma_d,2} = B^\sigma_{2,2}$ is a Hilbert-Sobolev space, for $\sigma \in \mathbb{R}$, then $d_{\mathcal{F}_d} = \|\cdot - \cdot\|_{\mathcal{W}^{-\sigma_d,2}}$ is the corresponding negative Sobolev pseudometric [57]. Recent work [27, 46, 28] established a minimax rate of $n^{-\frac{\sigma_g+\sigma_d}{2\sigma_g+1}} + n^{-1/2}$ when $\mathcal{F}_g = \mathcal{W}^{\sigma_g,2}$ is also a Hilbert-Sobolev space.

## 4 Main Results

The three **main technical contributions** of this paper are as follows:

1. We prove lower and upper bounds (Theorems 4 and 5, respectively) on minimax convergence rates of distribution estimation under IPM losses when the distribution class $\mathcal{P} = B^{\sigma_g}_{p_g,q_g}$ and the discriminator class $\mathcal{F} = B^{\sigma_d}_{p_d,q_d}$ are Besov spaces; these rates match up to polylogarithmic factors in the sample size $n$. Our upper bounds use the wavelet-thresholding estimator proposed in Donoho et al. [13], which we show converges at the optimal rate for a much wider range of losses than previously known. Specifically, if $M(\mathcal{F}, \mathcal{P})$ denotes minimax risk (2), we show that for $p'_d \geq p_g$, $\sigma_g \geq D/p_g$,

$$M\left(B_{p_d,q_d}^{\sigma_d}, B_{p_g,q_g}^{\sigma_g}\right) \asymp \max\left\{n^{-1/2}, n^{-\frac{\sigma_g+\sigma_d}{2\sigma_g+D}}, n^{-\frac{\sigma_g+\sigma_d+D(1-1/p_g-1/p_d)}{2\sigma_g+D(1-2/p_g)}}\right\}. \tag{3}$$

2. We show (Theorem 7) that, for $p_d' \geq p_g$ and $\sigma_g \geq D/p_g$, no estimator in a large class of distribution estimators, called "linear estimators", can converge at a rate faster than

$$M_{\mathrm{lin}}\left(B_{p_d,q_d}^{\sigma_d}, B_{p_g,q_g}^{\sigma_g}\right) \gtrsim n^{-\frac{\sigma_g+\sigma_d-D/p_g+D/p_d'}{2\sigma_g+D(1-2/p_g)+2D/p_d'}}. \tag{4}$$

"Linear estimators" include the empirical distribution, kernel density estimates with uniform bandwidth, and the orthogonal series estimators recently used in Liang [27] and Singh et al. [46]). The lower bound (4) implies that, in many settings (discussed in Section 5), linear estimators converge at sub-optimal rates. This effect is especially pronounced when the data dimension $D$ is large and the distribution $P$ has relatively sparse support (e.g., if $P$ is supported near a low-dimensional manifold). 3. We show that the minimax convergence rate can be achieved by a GAN with generator and discriminator networks of bounded size, after some regularization. As one of the first theoretical results separating performance of GANs from that of classic nonparametric tools such as kernel methods, this may help explain GANs' successes with high-dimensional data such as images.

## 4.1 Minimax Rates over Besov Spaces

We now present our main lower and upper bounds for estimating densities that live in a Besov space under a Besov IPM loss. Then, we have the following lower bound on the convergence rate:

**Theorem 4.** (***Lower Bound***) *Let* $r > \sigma_g \geq D/p_g$, *then,*

$$M\left(B_{p_d,q_d}^{\sigma_d}, B_{p_g,q_g}^{\sigma_g}\right) \gtrsim \max\left(n^{-\frac{\sigma_g+\sigma_d}{2\sigma_g+D}}, \left(\frac{\log n}{n}\right)^{\frac{\sigma_g+\sigma_d+D-D/p_g-D/p_d}{2\sigma_g+D-2D/p_g}}\right) \tag{5}$$

Before giving a corresponding upper bound, we describe the estimator on which it depends.

**Wavelet-Thresholding:** Our upper bound uses the wavelet-thresholding estimator proposed by [13]:

$$\widehat{p}_n = \sum_{k\in\mathbb{Z}} \widehat{\alpha}_k \phi_k + \sum_{j=0}^{j_0} \sum_{\lambda\in\Lambda_j} \widehat{\beta}_\lambda \psi_\lambda + \sum_{j=j_0}^{j_1} \sum_{\lambda\in\Lambda_j} \widetilde{\beta}_\lambda \psi_\lambda. \tag{6}$$

$\widehat{p}_n$ estimates $p$ via its truncated wavelet expansion, with $\widehat{\alpha}_k = \frac{1}{n}\sum_{i=1}^n \phi_k(X_i)$, $\widehat{\beta}_\lambda = \frac{1}{n}\sum_{i=1}^n \psi_\lambda(X_i)$, and $\widetilde{\beta}_\lambda = \widehat{\beta}_\lambda \mathbf{1}_{\{\widehat{\beta}_\lambda > \sqrt{j/n}\}}$ are empirical estimates of respective coefficient of the wavelet expansion of $p$. As [13] first showed, attaining optimality over Besov spaces requires truncating high-resolution terms (of order $j \in [j_0, j_1]$) when their empirical estimates are too small; this "nonlinear" part of the estimator distinguishes it from the "linear" estimators we study in the next section. The hyperparameters $j_0$ and $j_1$ are set to $j_0 = \frac{1}{2\sigma_g+D}\log_2 n$, $j_1 = \frac{1}{2\sigma_g+D-2D/p_g}\log_2 n$.

**Theorem 5.** (***Upper Bound***) *Let* $r > \sigma_g \geq D/p_g$ *and* $p_d' > p_g$. *Then, for a constant* $C$ *depending only on* $p_d'$, $\sigma_g$, $p_g$, $q_g$, $D$, $L_g$, $L_d$ *and* $\|\psi_\epsilon\|_{p_d'}$,

$$M\left(B_{p_d,q_d}^{\sigma_d}, B_{p_g,q_g}^{\sigma_g}\right) \leq C\left(\sqrt{\log n}\left(n^{-\frac{\sigma_g+\sigma_d}{2\sigma_g+D}} + n^{-\frac{\sigma_g+\sigma_d-D/p_g+D/p_d'}{2\sigma_g+D-2D/p_g}}\right) + n^{-1/2}\right) \tag{7}$$

We will comment only briefly on Theorems 4 and 5 here, leaving extended discussion for Section 5. First, note that the lower bound (5) and upper bound (7) are essentially tight; they differ only by a polylogarithmic factor in $n$. Second, both bounds contain two main terms of interest. The simpler term, $n^{-\frac{\sigma_g+\sigma_d}{2\sigma_g+D}}$, matches the rate observed in the Sobolev case by Singh et al. [46]. The other term is unique to more general Besov spaces. Depending on the values of $D, \sigma_d, \sigma_g, p_d,$ and $p_g$, one of these two terms dominates, leading to two main regimes of convergence rates, which we call the "Sparse" regime and the "Dense" regime. Section 5 discusses these and other interesting phenomena in detail.

## 4.2 Minimax Rates of Linear Estimators over Besov Spaces

We now show that, for many Besov densities and IPM losses, many widely-used nonparametric density estimators cannot converge at the optimal rate (5). These estimators are as follows:

**Definition 6** (Linear Estimator). Let $(\Omega, \mathcal{F}, P)$ be a probability space. An estimate $\widehat{P}$ of $P$ is said to be *linear* if there exist functions $T_i(X_i, \cdot) : \mathcal{F} \to \mathbb{R}$ such that for all measurable $A \in \mathcal{F}$,

$$\widehat{P}(A) = \sum_{i=1}^{n} T_i(X_i, A). \tag{8}$$

Classic examples of linear estimators include the empirical distribution ($T_i(X_i, A) = \frac{1}{n} 1_{\{X_i \in A\}}$, the kernel density estimate ($T_i(X_i, A) = \frac{1}{n} \int_A K(X_i, \cdot)$ for some bandwidth $h > 0$ and smoothing kernel $K : \mathcal{X} \times \mathcal{X} \to \mathbb{R}$) and the orthogonal series estimate ($T_i(X_i, A) = \frac{1}{n} \sum_{j=1}^{J} g_j(X_i) \int_A g_j$ for some cutoff $J$ and orthonormal basis $\{g_j\}_{j=1}^{\infty}$ (e.g., Fourier, wavelet, or polynomial) of $L^2(\Omega)$).

**Theorem 7** (Minimax rate for Linear Estimators). *Suppose* $r > \sigma_g \geq D/p_g$,

$$M_{lin}\left( B_{p_d, q_d}^{\sigma_d}, B_{p_g, q_g}^{\sigma_g} \right) := \inf_{\widehat{P}_{lin}} \sup_{p \in \mathcal{F}_g} \mathop{\mathbb{E}}_{X_{1:n}} \left[ d_{\mathcal{F}_d}\left( \mu_p, \widehat{P} \right) \right] \asymp n^{-\frac{1}{2}} + n^{-\frac{\sigma_g + \sigma_d - D/p_g + D/p_d'}{2\sigma_g + D - 2D/p_g + 2D/p_d'}} + n^{-\frac{\sigma_g + \sigma_d}{2\sigma_g + D}}$$

*where the* $\inf$ *is over all linear estimates of* $p \in \mathcal{F}_g$, *and* $\mu_p$ *is the distribution with density p.*

One can check that the above error decays no faster than $n^{-\frac{\sigma_g + \sigma_d + D - D/p_g - D/p_d}{2\sigma_g + D - 2D/p_g}}$. Comparing with the rate in Theorem 5, this implies that, in certain cases, convergence the rate for linear estimators is strictly slower than that for general estimators; i.e., linear estimators fail to achieve the minimax optimal rate over certain Besov space. We defer detailed discussion of this phenomenon to Section 5.

## 4.3 Upper Bounds on a Generative Adversarial Network

Pioneered by Goodfellow et al. [20] as a mechanism for applying deep neural networks to the problem of unsupervised image generation, Generative adversarial networks (GANs) have since been widely applied not only to computer vision [59, 24], but also to such diverse problems and data as machine translation using natural language data [56], discovering drugs [22] and designing materials [44] using molecular structure data, inferring expression levels using gene expression data [11], and sharing patient data under privacy constraints using electronic health records [8]. Besides the Jensen-Shannon divergence used by [20], many GAN formulations have been proposed based on minimizing other losses, including the Wasserstein metric [4, 21], total variation distance [30], $\chi^2$ divergence [32], MMD [26], Dudley metric [1], and Sobolev metric [37]. The diversity of data types and losses with which GANs have been used motivates studying GANs in a very general (nonparametric) setting. In particular, Besov spaces likely comprise the largest widely-studied family of nonparametric smoothness class; indeed, most of the losses listed above are Besov IPMs.

GANs are typically described as a two-player minimax game between a generator network $N_g$ and a discriminator network $N_d$; we denote by $\mathcal{F}_d$ the class of functions that can be implemented by $N_d$ and by $\mathcal{F}_g$ the class of distributions that can be implemented by $N_g$. A recent line of work has argued that a natural statistical model for a GAN as a distribution estimator is

$$\widehat{P} := \mathop{\mathrm{argmin}}_{Q \in \mathcal{F}_g} \sup_{f \in \mathcal{F}_d} \mathop{\mathbb{E}}_{X \sim Q} [f(X)] - \mathop{\mathbb{E}}_{X \sim \widetilde{P}_n} [f(X)], \tag{9}$$

where $\widetilde{P}_n$ is an (appropriately regularized) empirical distribution, and that, when $\mathcal{F}_d$ and $\mathcal{F}_g$ respectively approximate classes $\mathcal{F}$ and $\mathcal{P}$ well, one can bound the risk, under $\mathcal{F}$-IPM loss, of estimating distributions in $\mathcal{P}$ by (9) [31, 27, 46, 28]. We emphasize, that, as Singh et al. [46] showed, the minimax risk in this framework is identical to that under the "sampling" (or "implicit generative modeling" [36]) framework in terms of which GANs are usually cast. [4]

In this section, we show such a result for Besov spaces; namely, we show the existence of a particular GAN (specifically, a sequence of GANs, necessarily growing with the sample size $n$), that estimates distributions in a Besov space at the minimax optimal rate (7) under Besov IPM losses. This

construction uses a standard neural network architecture (a fully-connected neural network with rectified linear unit (ReLU) activations), and a simple data regularizer $\widetilde{P}_n$, namely the wavelet-thresholding estimator described in Section 4.1. Our results extend those of Liang [27] and Singh et al. [46], for Wasserstein loss over Sobolev spaces, to general Besov IPM losses over Besov spaces. We begin with a formal definition of the network architectures that we consider:

**Definition 8.** A *fully-connected ReLU network* $f_{(A_1,\ldots,A_H),(b_1,\ldots,b_H)} : \mathbb{R}^W \to \mathbb{R}$ has the form

$$A_H \eta\left(A_{H-1}\eta\left(\cdots \eta(A_1 x + b_1)\cdots\right) + b_{H-1}\right) + b_H,$$

where, for each $\ell \in [H-1]$, $A_\ell \in \mathbb{R}^{W \times W}$, and $A_H \in \mathbb{R}^{1 \times W}$ and the ReLU operation $\eta(x) = \max\{x, 0\}$ is applied element-wise to vectors in $\mathbb{R}^W$.

The size of $f_{(A_1,\ldots,A_H),(b_1,\ldots,b_H)}(x)$ can be measured in terms of the following four (hyper)parameters: the *depth $H$*, the *width $W$*, the *sparsity $S := \sum_{\ell\in[H]} \|A_\ell\|_{0,0} + \|b_\ell\|_0$* (i.e., the total number of non-zero weights), and the *maximum weight $B := \max\{\|A_\ell\|_{\infty,\infty}, \|b_\ell\|_\infty : \ell \in [H]\}$*. For given size parameters $H, W, S, B$ we write $\Phi(H, W, S, B)$ to denote the set of functions satisfying the corresponding size constraints.

Our results rely on a recent construction (Lemma 17 in the Appendix), by [49], of a fully-connected ReLU network that approximates Besov functions. [49] used this approximation to bound the risk of a neural network for nonparametric regression over Besov spaces, under $L^r$ loss. Here, we use this approximation result Lemma 17 to bound the risk of a GAN for nonparametric distribution estimation over Besov spaces, under the much larger class of Besov IPM losses. Our precise result is as follows:

**Theorem 9** (Convergence Rate of a Well-Optimized GAN)**.** *Fix a Besov density class $B_{p_g,q_g}^{\sigma_g}$ with $\sigma_g > D/p_g$ and discriminator class $B_{p_d,q_d}^{\sigma_d}$ with $\sigma_d > D/p_d$. Then, for any desired approximation error $\epsilon > 0$, one can construct a GAN $\widehat{p}$ of the form (9) (with $\widetilde{p}_n$) with discriminator network $N_d \in \Phi(H_d, W_d, S_d, B_d)$ and generator network $N_g \in \Phi(H_g, W_g, S_g, B_g)$, s.t. for all $p \in B_{p_g,q_g}^{\sigma_g}$*

$$\mathbb{E}\left[d_{B_{p_d,q_d}^{\sigma_d}}(\widehat{p}, p)\right] \lesssim \epsilon + \mathbb{E}\, d_{B_{p_d,q_d}^{\sigma_d}}(\widetilde{p}_n, p)$$

*where $H_d$, $H_g$ grow logarithmically with $1/\epsilon$, $W_d, S_d, B_d, W_g, S_g, B_g$ grow polynomially with $1/\epsilon$ and $C > 0$ is a constant that depends only on $B_{p_d,q_d}^{\sigma_d}$ and $B_{p_g,q_g}^{\sigma_g}$.*

This theorem implies that the rate of convergence of the GAN estimate $\widehat{p}$ of the form 9 is the same as the convergence rate of the estimator $\widetilde{p}_n$ with which the GAN estimate is generated (Here we assume that all distributions have densities). Therefore, given our upper bound from theorem 5 we have the following direct consequence.

**Corollary 10.** *For a Besov density class $B_{p_g,q_g}^{\sigma_g}$ with $\sigma_g > D/p_g$ and discriminator class $B_{p_d,q_d}^{\sigma_d}$ with $\sigma_d > D/p_d$ there exists an appropriately constructed GAN estimate $\widehat{p}$ s.t.*

$$d_{\mathcal{F}_d}(\widehat{p}, p) \leq \left(n^{-\eta(D,\sigma_d,p_d,\sigma_g,p_g)}\sqrt{\log n}\right)$$

*where $\eta(D, \sigma_d, p_d, \sigma_g, p_g) = \min\left\{\frac{1}{2}, \frac{\sigma_g+\sigma_d}{2\sigma_g+D}, \frac{\sigma_g+\sigma_d+D-D/p_g-D/p_d'}{2\sigma_g+D(1-2/p_g)}\right\}$ is the exponent from (7).*

In other words there is a GAN estimate that is minimax rate optimal for a smooth class of densities over an IPM generated by a smooth class of discriminator functions.

# 5 Discussion of Results

In this section, we discuss some general phenomena that can be gleaned from our technical results.

First, we note that, perhaps surprisingly, $q_d$ and $q_g$ do not appear in our bounds. Tao [50] suggests that $q_d$ and $q_g$ may have only logarithmic effects (contrasted with the polynomial effects of $\sigma_d$, $p_d$, $\sigma_g$, and $p_g$). Thus, a more fine-grained analysis to close the polylogarithmic gap between our lower and upper bounds for general estimators (Theorems 4 and 5) might require incorporating $q_d$ and $q_g$.

On the other hand, the parameters $\sigma_d$, $p_d$, $\sigma_g$, and $p_g$ each play a significant role in determining minimax convergence rates, in both the linear and general cases. We first discuss each of these parameters independently, and then discuss some interactions between them.

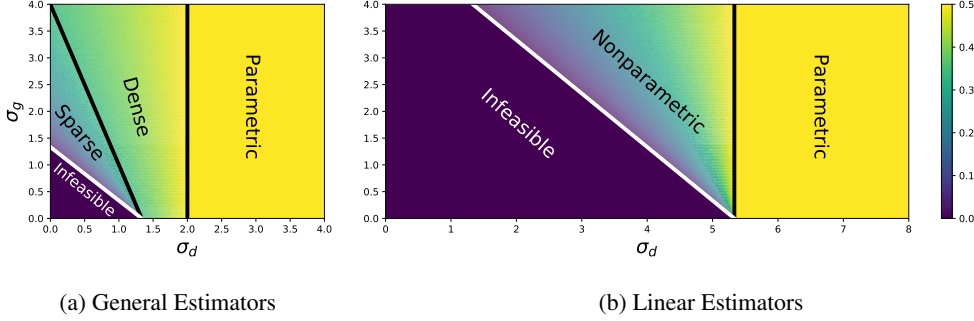

(a) General Estimators                    (b) Linear Estimators

Figure 1: Minimax convergence rates as functions of discriminator smoothness $\sigma_d$ and distribution function smoothness $\sigma_g$, for (a) general and (b) linear estimators, in the case $D = 4$, $p_d = 1.2$, $p_g = 2$. Color shows exponent of minimax convergence rate (i.e., $\alpha(\sigma_d, \sigma_g)$ such that $M\left(B_{1.2,q_d}^{\sigma_d}(\mathbb{R}^D), B_{2,q_g}^{\sigma_g}(\mathbb{R}^D)\right) \asymp n^{-\alpha(\sigma_d,\sigma_g)}$), ignoring polylogarithmic factors.

**Roles of the smoothness orders $\sigma_d$ and $\sigma_g$**    As a visual aid for understanding our results, Figure 1 show phase diagrams of minimax convergence rates, as functions of discriminator smoothness $\sigma_d$ and distribution smoothness $\sigma_g$, in the illustrative case $D = 4$, $p_d = 1.2$, $p_g = 2$. When $1/p_g + 1/p_d > 1$, a minimum total smoothness $\sigma_d + \sigma_g \geq D(1/p_d + 1/p_g - 1)$ is needed for consistent estimation to be possible – this fails in the "Infeasible" region of the phase diagrams. Intuitively, this occurs because $\mathcal{F}_d$ is not contained in the topological dual $\mathcal{F}'_g$ of $\mathcal{F}_g$. For linear estimators, even greater smoothness $\sigma_d + \sigma_g \geq D(1/p_d + 1/p_g)$ is needed. At the other extreme, for highly smooth discriminator functions, both linear and nonlinear estimators converge at the parametric rate $O\left(n^{-1/2}\right)$, corresponding to the "Parametric" region. In between, rates for linear estimators vary smoothly with $\sigma_d$ and $\sigma_g$, while rates for nonlinear estimators exhibit another phase transition on the line $\sigma_g + 3\sigma_d = D$; to the left lies the "Sparse" case, in which estimation error is dominated by a small number of large errors at locations where the distribution exhibits high local variation; to the right lies the "Dense" case, where error is relatively uniform on the sample space.

The left boundary $\sigma_d = 0$ corresponds to the classical results of Donoho et al. [13], who consequently identified the "Infeasible", "Sparse", and "Dense" phases, but not the "Parametric" phase. When restricting to linear estimators, the "Infeasible" region grows and the "Parametric" region shrinks.

**Role of the powers $p_d$ and $p_g$**    At one extreme ($p_d = \infty$) lie $L^1$ or total variation loss ($\sigma_d = 0$), Wasserstein loss ($\sigma_d = 1$), and its higher-order generalizations, for which we showed the rate

$$M\left(B_{\infty,q_d}^{\sigma_d}, B_{p_g,q_g}^{\sigma_g}\right) \asymp n^{-\frac{\sigma_g+\sigma_d}{2\sigma_g+D}} + n^{-1/2},$$

generalizing the rate first shown by Singh et al. [46] for Hilbert-Sobolev classes to other distribution classes, such as $\mathcal{F}_g = $ BV. Because discriminator functions in this class exhibit homogeneous smoothness, these losses effectively weight the sample space relatively uniformly in importance, the "Sparse" region in Figure (1a) vanishes, and linear estimators can perform optimally.

At the other extreme ($p_d = 1$) lie $L^\infty$ loss ($\sigma_d = 0$), Kolmogorov-Smirnov loss ($\sigma_d = 1$), and its higher-order generalizations, for which we have shown that the rate is always

$$M\left(B_{1,q_d}^{\sigma_d}, B_{p_g,q_g}^{\sigma_g}\right) \asymp n^{-\frac{\sigma_g+\sigma_d+D(1-1/p_d-1/p_g)}{2\sigma_g+D(1-2/p_g)}} + n^{-1/2};$$

except in the parametric regime ($D \leq 2\sigma_d$), this rate differs from that of Singh et al. [46]. Because discriminator functions can have inhomogeneous smoothness, and hence weight some portions of the sample space much more heavily than others, the "Dense" region in Figure 1a vanishes, and linear estimators are always sub-optimal. We note that Sadhanala et al. [43] recently proposed using these higher-order distances (integer $\sigma_d > 1$) in a fast two-sample test that generalizes the well-known Kolmogorov-Smirnov test, improving sensitivity to the tails of distributions; our results may provide a step towards understanding theoretical properties of this test.

**Comparison of linear and general rates**    Letting $\sigma'_g := \sigma_g - D(1/p_g + 1/p_d)$, one can write the sparse term of the linear minimax rate in the same form as the Dense rate, replacing $\sigma_g$ with $\sigma'_g$:

$$M_{\text{lin}}\left(B_{p_d,q_d}^{\sigma_d}, B_{p_g,q_g}^{\sigma_g}\right) \asymp n^{-\frac{\sigma_g'+\sigma_d}{2\sigma_g'+D}}. \tag{10}$$

This is not a coincidence; Morrey's inequality [17, Section 5.6.2] in functional analysis tells us that for general $\sigma_g > D(1/p_g + 1/p_d)$, $\sigma_g' := \sigma_g - D(1/p_g + 1/p_d)$ is largest possible value such that the embedding $B_{p_g,p_g}^{\sigma_g} \subseteq B_{p_d,p_d}^{\sigma_g'}$ holds. In the extreme case $p_d = \infty$ (corresponding to generalizations of total variation loss), one can interpret the rate (10) as saying that linear estimators benefit only from homogeneous (e.g., Hölder) smoothness, and not from weaker inhomogeneous (e.g., Besov) smoothness. For general $p_d$, linear estimator can still benefit from inhomogeneous smoothness, but to a lesser extent than general minimax optimal estimators.

**Conclusions**   We have shown, up to log factors, unified minimax convergence rates for a large class of pairs of $\mathcal{F}_d$-IPM losses and distribution classes $\mathcal{F}_g$. By doing so, we have generalized several phenomena that had observed in special cases previously. First, under sufficiently weak loss functions, distribution estimation is possible at the parametric rate $O(n^{-1/2})$ even over very large nonparametric distribution classes. Second, in many cases, optimal estimation requires estimators that adapt to inhomogeneous smoothness conditions; many commonly used distribution estimators fail to do this, and hence converge at sub-optimal rates, or even fail to converge. Finally, GANs with sufficiently large fully-connected ReLU neural networks using wavelet-thresholding regularization perform statistically minimax rate-optimal distribution estimation over inhomogeneous nonparametric smoothness classes (assuming the GAN optimization problem can be solved accurately). Importantly, since GANs optimize IPM losses much weaker than traditional $L^p$ losses, they may be able to learn reasonable approximations of even high-dimensional distributions with tractable sample complexity, perhaps explaining why they excel in the case of image data. Thus, our results suggest that the curse of dimensionality may be less severe than indicated by classical nonparametric lower bounds.

## Footnotes

[3]We assume a good optimization algorithm for computing (1), although this is also an active area of research.

[4]As in these previous works, we assume implicitly that the optimum (9) can be computed; this complex saddle-point problem is itself the subject of a related but distinct and highly active area of work [39, 3, 29, 19].

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
