[Supplementary Material · NeurIPS19_Besov.pdf]

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

## A  Technical Definitions and Notation

As noted in the main text, we need a multiresolution approximation (MRA) satisfying an $r$-regularity condition, defined as follows:

**Definition 11.** Given a non-negative integer $r$, an MRA is called *r-regular* if the function $\phi$ can be chosen in such a way that, for every $m \in \mathbb{N}$ and multi-index $\alpha = (\alpha_1, \ldots, \alpha_D) \in \mathbb{N}^D$ satisfying $|\alpha| \le r$, for some constant $C_{\alpha,m}$, $|\partial^\alpha \phi(x)| \le C_{\alpha,m}(1 + |x|)^{-m}$. Here, $\partial^\alpha = (\partial/\partial x_1)^{\alpha_1} \cdots (\partial/\partial x_D)^{\alpha_D}$ is the mixed derivative of index $\alpha$, $|\alpha| = \sum_{j=1}^D \alpha_j$ and $|x|$ is any of the equivalent norms on a finite dimensional Euclidean space. That is, all derivatives of $\phi$ of order up to $r$ are bounded and decay at a rate faster than any polynomial.

While constructing an $r$-regular MRA is nontrivial, it suffices for our purpose to note that $r$-regular MRAs exist; the most famous example is the Daubechies wavelet [11, 36].

We also note the following result showing that for any function in $V_j$ (i.e., at a certain "level" in the MRA) its $L^p$ norm is equivalent to the $l^p$ sequence norm of its coefficients in the wavelet basis; this helps motivate the sequence-based definition of the Besov norm.

**Proposition 12** (Meyer [36], Section 6.10, Proposition 7)**.** *There exist positive constants $C, C'$ s.t. for every $1 \le p \le \infty$, $j \in \mathbb{Z}$ and $\{\alpha_k\} \in l^p$, $f(x) = \sum a_k 2^{Dj/2} \psi_\epsilon(2^j x - k)$, $\epsilon \in E, k \in \mathbb{Z}^D$,*

$$C \|f\|_p \le 2^{Dj(1/2 - 1/p)} \left(\sum |a_k|^p\right)^{1/p} \le C' \|f\|_p.$$

Appendix A.1 of Donoho et al. [14] offers a more extended background of Besov spaces, including how the sequence-based definition corresponds to more conventional smoothness measures (moduli of continuity), as well as some direct connections between Besov spaces and minimax theory for linear estimators.

# B    Upper Bound - Linear Case

For any density function $p$ let

$$\alpha_k^p = \int \phi_k(x) p(x) dx$$

$$\beta_\lambda^p = \int \psi_\lambda(x) p(x) dx$$

We first show that Besov IPMs essentially measure the distance in co-efficient space between compactly supported densities.

**Lemma 13.** *For any compactly supported probability densities $p$, $q \in L_{p'_d}$ where $\mathcal{F}_d = B_{p_d, q_d}^{\sigma_d}$*

$$d_{\mathcal{F}_d}(p, q) = \sup_{f \in \mathcal{F}_d} \left| \sum_{k \in \mathbb{Z}} \alpha_k^f (\alpha_k^p - \alpha_k^q) + \sum_{j \geq 0} \sum_{\lambda \in \Lambda} \beta_\lambda^f (\beta_\lambda^p - \beta_\lambda^q) \right|$$

*where for $f \in \mathcal{F}_d$*

$$f = \sum_{k \in \mathbb{Z}} \alpha_k^f \phi_k + \sum_{j \geq 0} \sum_{\lambda \in \Lambda_j} \beta_\lambda^f \psi_\lambda$$

*Proof.* We notice that the convergence to $f$ above is in the $L_\infty$ norm. So for probability measures $P, Q$ we have,

$$d_{\mathcal{F}_d}(p, q) = \sup_{f \in \mathcal{F}_d} |\mathrm{E}_{X \sim p}[f(X)] - \mathrm{E}_{X \sim q}[f(X)]|$$

$$= \sup_{f \in \mathcal{F}_d} \left| \int_\mathcal{X} f(x) p(x) dx - f(x) q(x) dx \right|$$

$$= \sup_{f \in \mathcal{F}_d} \left| \int_\mathcal{X} \left( \sum_{k \in \mathbb{Z}} \alpha_k^f \phi_k(x) + \sum_{j \geq 0} \sum_{\lambda \in \Lambda_j} \beta_\lambda^f \psi_\lambda(x) \right) (p(x) - q(x)) dx \right|$$

If $p, q$ are compactly supported on $[-B, B]$ then we can assume WLOG that $f$ is compactly supported on $[-B, B]$ so convergence of $f_n$ to $f$ in $L^\infty$ norm implies convergence in $L^1$ norm. Therefore,

$$d_{\mathcal{F}_d}(P, Q) = \sup_{f \in \mathcal{F}_d} \left| \sum_{k \in \mathbb{Z}} \int_\mathcal{X} \alpha_k^f \phi_k (dP(x) - dQ(x)) + \sum_{j \geq 0} \sum_{\lambda \in \Lambda_j} \int_\mathcal{X} \beta_\lambda^f \psi_\lambda (dP(x) - dQ(x)) \right|$$

$$= \sup_{f \in \mathcal{F}_d} \left| \sum_{k \in \mathbb{Z}} \alpha_k^f (\alpha_k^p - \alpha_k^q) + \sum_{j \geq 0} \sum_{\lambda \in \Lambda} \beta_\lambda^f (\beta_\lambda^p - \beta_\lambda^q) \right|$$

$\square$

We will need the following inequalities to estimate the error of the wavelet estimator under the IPM loss.

The first lemma is the standard upper bound on the $m$th moment of a sum of IID random variables with bounded variance. The second is a standard concentration inequality used to bound large deviations in our error estimate.

**Lemma 14.** *(**Rosenthal's Inequality** ([44])) Let $m \in \mathbb{R}$ and $Y_1, \ldots, Y_n$ be IID random variables with $\mathbb{E}[Y_i] = 0$, $\mathbb{E}[Y_i^2] \leq \sigma^2$. Then there is a constant $c_m$ that depends only on $m$ s.t.*

$$\mathbb{E}\left[ \left| \frac{1}{n} \sum_{i=1}^n Y_i \right|^m \right] \leq c_m \left( \frac{\sigma^m}{n^{m/2}} + \frac{\mathbb{E}|Y_1|^m}{n^{m-1}} \right) \qquad \text{for } 2 < m < \infty,$$

$$\mathbb{E}\left[ \left| \frac{1}{n} \sum_{i=1}^n Y_i \right|^m \right] \leq \sigma^m n^{-m/2} \qquad \text{for } 1 \leq m \leq 2.$$

**Lemma 15.** *(**Bernstein's Inequality** ([6])) If $Y_1, \ldots, Y_n$ are IID random variables such that $\mathbb{E}[Y_i] = 0$, $\mathbb{E}[Y_i^2] = \sigma^2$ and $|Y_i| \leq \|Y\|_\infty < \infty$, then*

$$\Pr\left(\left|\frac{1}{n}\sum_{i=1}^n Y_i\right| > \lambda\right) \leq 2\exp\left(-\frac{n\lambda^2}{2(\sigma^2 + \|Y\|_\infty \lambda/3)}\right)$$

*where $\|Y\|_\infty = \operatorname{ess\,sup} Y$.*

Given discriminator and generator classes as

$$\mathcal{F}_d = \{f : \|f\|_{p_d,q_d}^{\sigma_d} \leq L_d\}$$
$$\mathcal{F}_g = \{p : \|p\|_{p_g,q_g}^{\sigma_g} \leq L_g\} \cap \mathcal{P}$$
$$\mathcal{P} = \{p : p \geq 0, \|p\|_{L^1} = 1, \operatorname{supp}(p) \subseteq [-T,T]\},$$

we decompose $f \in \mathcal{F}_d$ as

$$f = \sum_{k\in\mathbb{Z}} \alpha_k \phi_k + \sum_{j\geq 0}\sum_{\lambda\in\Lambda_j} \beta_\lambda \psi_\lambda.$$

We use the linear wavelet estimator to demonstrate the upper bound. Let $X_1, \ldots, X_n$ be IID with density $p \in \mathcal{F}_g$ and consider the wavelet estimator of $p$ i.e.

$$p = \sum_{k\in\mathbb{Z}} \alpha_k^p \phi_k + \sum_{j\geq 0}\sum_{\lambda\in\Lambda_j} \beta_\lambda^p \psi_\lambda$$

$$\widehat{p}_n = \sum_{k\in\mathbb{Z}} \widehat{\alpha}_k \phi_k + \sum_{j=0}^{j_0}\sum_{\lambda\in\Lambda_j} \widehat{\beta}_\lambda \psi_\lambda$$

where

$$\alpha_k^p = \mathop{\mathbb{E}}_{X\sim p}[\phi_k(X)] \qquad \widehat{\alpha}_k = \frac{1}{n}\sum_{i=1}^n \phi_k(X_i)$$
$$\beta_\lambda^p = \mathop{\mathbb{E}}_{X\sim p}[\psi_\lambda(X)] \qquad \widehat{\beta}_\lambda = \frac{1}{n}\sum_{i=1}^n \psi_\lambda(X_i)$$

Then applying lemma 13, we bound

$$d_{\mathcal{F}_d}(p,\widehat{p}_n) \leq \sup_{f\in\mathcal{F}_d}\sum_{k\in\mathbb{Z}} \alpha_k\left(\alpha_k^p - \widehat{\alpha}_k\right) \qquad + \sup_{f\in\mathcal{F}_d}\sum_{j=0}^{j_0}\sum_{\lambda\in\Lambda_j} \beta_\lambda\left(\beta_\lambda^p - \widehat{\beta}_\lambda\right)$$
$$+ \sup_{f\in\mathcal{F}_d}\sum_{j\geq j_1}\sum_{\lambda\in\Lambda_j} \beta_\lambda \beta_\lambda^p$$

where the first two terms constitute the stochastic error and the last term is the bias. We bound these separately below. We first prove a few lemmas that will be used repeatedly to upper bound the different terms.

**Lemma 16.** *Let $n_1, n_2 \in \mathbb{N} \cup \{\infty\}$ and $\eta$ be any sequence of numbers. Then*

$$\mathop{\mathbb{E}}_{X_1,\ldots,X_n}\sup_{f\in\mathcal{F}_d}\sum_{j=n_1}^{n_2}\sum_{\lambda\in\Lambda_j} \gamma_\lambda \eta_\lambda \leq L_D \sum_{j=n_1}^{n_2} 2^{-j(\sigma_d + D/2 - D/p_d)}\left(\mathop{\mathbb{E}}_{X_1,\ldots,X_n}\sum_{\lambda\in\Lambda_j} |\eta_\lambda|^{p_d'}\right)^{1/p_d'}$$

*Note that if the above is true also if $\gamma = \alpha^f$ and $n_1 = n_2 = 0$.*

*Proof.* Since $f \in \mathcal{F}_d$, applying Hölder's inequality twice we get,

$$\mathbb{E}_{X_1,\dots,X_n} \sup_{f \in \mathcal{F}_d} \sum_{j=n_1}^{n_2} \sum_{\lambda \in \Lambda_j} \gamma_\lambda \eta_\lambda \leq \mathbb{E}_{X_1,\dots,X_n} \sup_{f \in \mathcal{F}_d} \sum_{j=n_1}^{n_2} \|\gamma\|_{p_d} \|\eta\|_{p_d'}$$

$$\leq \mathbb{E}_{X_1,\dots,X_n} \sup_{f \in \mathcal{F}_d} \left( \sum_{j=n_1}^{n_2} \left( 2^{j(\sigma_d + D/2 - D/p_d)} \|\gamma\|_{p_d} \right)^{q_d} \right)^{1/q_d}$$

$$\times \sum_{j=n_1}^{n_2} 2^{-j(\sigma_d + D/2 - D/p_d)} \|\eta\|_{p_d'} \quad (l^1 \subseteq l^{q_d'})$$

$$\leq L_D \sum_{j=n_1}^{n_2} 2^{-j(\sigma_d + D/2 - D/p_d)} \mathbb{E}_{X_1,\dots,X_n} \|\eta\|_{p_d'}$$

$$\leq L_D \sum_{j=n_1}^{n_2} 2^{-j(\sigma_d + D/2 - D/p_d)} \left( \mathbb{E}_{X_1,\dots,X_n} \sum_{\lambda \in \Lambda_j} |\eta_\lambda|^{p_d'} \right)^{1/p_d'}$$

where $p_d'$ is the conjugate of $p_d$ i.e. $\frac{1}{p_d} + \frac{1}{p_d'} = 1$ and we applied Jensen's to get the last inequality. $\qquad \square$

**Lemma 17.** *Let $f \in B_{p_g, q_g}^{\sigma_g}$ where $\sigma_g > D/p_g$ then*

$$\|f\|_\infty \leq 4A \|\psi\|_\infty L_g (1 - 2^{(\sigma_g - D/p_g)q_g'})^{-1/q_g'}$$

*This implies that sufficiently smooth Besov spaces $B_{p_g, q_q}^{\sigma_g}$ are uniformly bounded.*

*Proof.* We have that $\sum_{k \in \mathbb{Z}^D} \alpha_k \phi_k + \sum_{j \geq 0} \sum_{\lambda \in \Lambda_j} \beta_\lambda \psi_\lambda$ converges to $f$ in $L_\infty$. So, using the fact that $l^{p_d} \subseteq l^\infty$ and proposition 12,

$$\|f\|_\infty \leq 2A \|\psi\|_\infty \left( \|\{\alpha_k\}_{k \in \mathbb{Z}^D}\|_\infty + \sum_{j \geq 0} 2^{Dj/2} \|\{\beta_\lambda\}_{\lambda \in \Lambda_j}\|_\infty \right).$$

We can upper bound, by Hölder's inequality,

$$\sum_{j \geq 0} 2^{Dj/2} \|\{\beta_\lambda\}_{\lambda \in \Lambda_j}\|_\infty \leq \sum_{j \geq 0} \frac{1}{2^{j(\sigma_g - D/p_g)}} \times 2^{j(\sigma_g + D/2 - D/p_g)} \|\{\beta_\lambda\}_{\lambda \in \Lambda_j}\|_\infty$$

$$\leq \left( \sum_{j \geq 0} \frac{1}{2^{j(\sigma_g - D/p_g)q_g'}} \right)^{1/q_g'} \left( \sum_{j \geq 0} 2^{jq_g(\sigma_g + D/2 - D/p_g)} \|\{\beta_\lambda\}_{\lambda \in \Lambda_j}\|_\infty^{q_g} \right)^{1/q_g}$$

$$\leq \left( \frac{1}{1 - 2^{-(\sigma_g - D/p_g)q_g'}} \right)^{1/q_g'} \left( \sum_{j \geq 0} 2^{jq_g(\sigma_g + D/2 - D/p_g)} \|\{\beta_\lambda\}_{\lambda \in \Lambda_j}\|_{p_g}^{q_g} \right)^{1/q_g}$$

$$\leq \left( 1 - 2^{-(\sigma_g - D/p_g)q_g'} \right)^{-1/q_g'} \|f\|_{p_g q_g}^{\sigma_g}$$

$$\leq \left( 1 - 2^{-(\sigma_g - D/p_g)q_g'} \right)^{-1/q_g'} L_g.$$

Putting the above together we obtain the required upper bound. $\qquad \square$

We also need a few preliminary results namely, the moments of error of linear estimates of the wavelet coefficients are essentially bounded by $1/\sqrt{n}$ and the probability that this error is large is negligibly small. In particular,

**Lemma 18.** *(Moment Bounds) Let* $X_1, \ldots, X_n \sim p$, $m \geq 1$ *s.t. there is a constant* $c$ *with* $\mathbb{E}_p |\psi_\lambda(X)|^m \leq c 2^{Dj(m/2-1)}$. *Let*

$$\gamma_\lambda^p = \mathbb{E}[\psi_\lambda(X)],$$

$$\widehat{\gamma}_\lambda = \frac{1}{n} \sum_{i=1}^n \psi_\lambda(X_i),$$

*Then for all* $j$ *s.t.* $2^{Dj} \in \mathcal{O}(n)$,

$$\mathbb{E}[|\widehat{\gamma}_{jk} - \gamma_{jk}|^m] \leq c n^{-m/2}.$$

*where* $c = c_m \left( \mathbb{E}_p |\psi_\lambda(X)|^2 \right)^{m/2}$ *is a constant.*

*Proof.* Since $\psi_\lambda$ is bounded for every $\lambda$, let

$$Y_i = \psi_\lambda(X_i) - \mathbb{E}[\psi_\lambda(X)]$$

then for all $m \geq 1$, applying Jensen's inequality repeatedly we get

$$\mathbb{E}[|Y_i|^m] \leq \mathbb{E}[(|\psi_\lambda(X_i)| + |\mathbb{E}[\psi_\lambda(X_i)]|)^m] \qquad \text{(triangle inequality)}$$

$$\leq 2^{m-1} \left( \mathbb{E}[|\psi_\lambda(X_i)|^m] + |\mathbb{E}[\psi_\lambda(X_i)]|^m \right) \qquad \text{(Jensen's)}$$

$$\leq 2^m \mathbb{E}[|\psi_\lambda(X_i)|^m]. \qquad \text{(Jensen's)}$$

Therefore, by Rosenthal's inequality we have,

$$\mathbb{E}[|\gamma_\lambda^p - \widehat{\gamma}_\lambda|^m] \leq c_m \left( \left( \mathbb{E}_p |\psi_\lambda(X)|^2 \right)^{m/2} + c \left( \frac{2^{Dj}}{n} \right)^{(m/2-1)_+} \right) n^{-m/2}$$

where $c_m$ is a constant that only depends on $m$. Therefore,

$$\mathbb{E}[|\gamma_\lambda^p - \widehat{\gamma}_\lambda|^m] \leq c_m \left( \mathbb{E}_p |\psi_\lambda(X)|^2 \right)^{m/2} n^{-m/2}$$

$\square$

Note that we have from above $2^{Dj_1} \leq n$ so this bound holds for any $j \leq j_1$.

**Lemma 19.** *(Large Deviations) Let* $X_1, \ldots, X_n \sim p$ *such that for a constant* $c$, $\mathbb{E}_p |\psi_\lambda(X)|^2 \leq c$. *Let*

$$\gamma_\lambda^p = \mathbb{E}[\psi_\lambda(X)],$$

$$\widehat{\gamma}_\lambda = \frac{1}{n} \sum_{i=1}^n \psi_\lambda(X_i),$$

*Let* $l = \sqrt{j/n}$ *and* $\gamma > 0$, *then, for all* $j$ *s.t.* $2^{Dj} \in o(n)$, *we have,*

$$\Pr(|\widehat{\gamma}_\lambda - \gamma_\lambda| > (K/2)l) \leq 2 \times 2^{-\gamma n l^2}$$

*where* $K$ *large enough such that*

$$\frac{K^2}{8(c + \|\psi_\epsilon\|_\infty (K/3))} > \log 2\gamma$$

*Proof.* Applying Bernstein's inequality we have

$$\Pr(|\widehat{\gamma}_\lambda - \gamma_\lambda| > (K/2)l) \leq 2 \exp\left( -\frac{n(K/2)^2 l^2}{2(c + 2^{Dj/2} \|\psi_\epsilon\|_\infty (K/3)l)} \right)$$

$$\leq 2 \exp\left( -\frac{K^2 n l^2}{8(L_g + \|\psi_\epsilon\|_\infty (K/3))} \right)$$

This implies for $K$ satisfying the above condition,

$$\Pr(|\widehat{\gamma}_\lambda - \gamma_\lambda| > (K/2)l) \leq 2 \times 2^{(-\gamma n l^2)}$$

$\square$

Now for every $j \le j_1$, $l$ satisfies the requirements of the above lemma. So if $nl^2(=j) \to \infty$ as $n \to \infty$ the probability of large deviation goes to zero.

**Lemma 20.** *(Variance) Let $X_1, \dots, X_n \sim p$ where $p$ is compactly supported, such that for a constant $c$, $\mathbb{E}_p |\psi_\lambda(X)|^m \le c2^{Dj(m/2-1)}$. Let $\mathcal{F}_d = B^{\sigma_d}_{p_d, q_d}$, then the variance of a linear wavelet estimator $\widehat{p}$ with $j_0$ terms i.e.*

$$\widehat{p}_n = \sum_{k \in \mathbb{Z}} \widehat{\alpha}_k \phi_k + \sum_{j=0}^{j_0} \sum_{\lambda \in \Lambda_j} \widehat{\beta}_\lambda \psi_\lambda$$

*is bounded by*

$$d_{\mathcal{F}_d}(\widehat{p}_n, \mathbb{E}[\widehat{p}_n]) \le c \left( \frac{1}{\sqrt{n}} + \frac{2^{j_0(D/2-\sigma_d)}}{\sqrt{n}} \right)$$

*where $c = c_{p'_d} \left( \mathbb{E}_p |\psi_\lambda(X)|^2 \right)^{1/2}$ is a constant.*

*Proof.* Since $\mathcal{F}_d = B^{\sigma_d}_{p_d, q_d}$ and $p$ is compactly supported we can, by lemma 13 upper bound

$$\mathbb{E}_{X_1, \dots, X_n} \sup_{f \in \mathcal{F}_d} \sum_{k \in \mathbb{Z}} \alpha_k^f (\alpha_k^p - \widehat{\alpha}_k) + \mathbb{E}_{X_1, \dots, X_n} \sup_{f \in \mathcal{F}_d} \sum_{j=0}^{j_0} \sum_{\lambda \in \Lambda_j} \beta_\lambda^f \left( \beta_\lambda^p - \widehat{\beta}_\lambda \right)$$

Since, for a constant $c$, $\mathbb{E}_p |\psi_\lambda(X)|^m \le c2^{Dj(m/2-1)}$ we can apply the moment bound below. For the first term we have, (taking $\gamma = \alpha$ and $n_1 = n_2 = 0$ in lemma 16 above)

$$\mathbb{E}_{X_1, \dots, X_n} \sup_{f \in \mathcal{F}_d} \sum_{k \in \mathbb{Z}} \alpha_k^f (\alpha_k^p - \widehat{\alpha}_k)$$

$$\le L_D \left( \sum_k \mathbb{E}_{X_1, \dots, X_n} |\alpha_k^p - \widehat{\alpha}_k|^{p'_d} \right)^{1/p'_d} \qquad \text{(finitely many terms)}$$

$$\le c L_D \|p\|_\infty \left( (T+A) n^{-p'_d/2} \right)^{1/p'_d} \qquad \text{(moment bound)}$$

$$\le c n^{-1/2}$$

where we use the fact only finitely many of the $\alpha$s are non-zero because of the compactness of the support of the densities we consider and the compactness of the wavelets. Similarly taking $\gamma = \beta$, $n_1 = 0$, $n_2 = j_0$ in lemms 16 we have, using the moment bound as above,

$$\mathbb{E}_{X_1, \dots, X_n} \sup_{f \in \mathcal{F}_d} \sum_{j=0}^{j_0} \sum_{\lambda \in \Lambda_j} \beta_\lambda^f \left( \beta_\lambda^p - \widehat{\beta}_\lambda \right)$$

$$\le c \|p\|_\infty L_D \sum_{j=0}^{j_0} 2^{-j(\sigma_d + D/2 - D/p_d)} \left( 2^{Dj}(T+A) n^{-p'_d/2} \right)^{1/p'_d}$$

$$\le L_D \sum_{j=0}^{j_0} 2^{-j(\sigma_d + D/2 - D/p_d)} 2^{Dj/p'_d} n^{-1/2}$$

$$\le c L_D \|p\|_\infty \sum_{j=0}^{j_0} 2^{j(D/2 - \sigma_d)} n^{-1/2}$$

$$\le c \|p\|_\infty \begin{cases} 2^{j_0(D/2-\sigma_d)} n^{-1/2} & \sigma_d \le D/2 \\ n^{-1/2} & \sigma_d > D/2 \end{cases}$$

$\square$

**Lemma 21.** *(Bias) Let $X_1, \dots, X_n \sim p$ where $p \in B^{\sigma_g}_{p_g, q_g}$ is compactly supported and $\sigma_g \ge D/p_g$, $\mathcal{F}_d = B^{\sigma_d}_{p_d, q_d}$. Then the bias of a linear wavelet estimator $\widehat{p}$ with $j_0$ terms is bounded by*

$$d_{\mathcal{F}_d}(p, \mathbb{E}[\widehat{p}_n]) \le c2^{-j_0(\sigma_d + \sigma_g - (D/p_g - D/p'_d)_+)}$$

*where $c$ is a constant that depends on $p_d$ and $\|\psi\|_m$.*

*Proof.* Since $p$ is compactly supported, by lemma 13 we need to upper bound

$$\sup_{\beta \in \mathcal{F}_d} \sum_{j \geq j_1} \sum_{\lambda \in \Lambda} \beta^f_\lambda \beta^p_\lambda$$

Using lemma 16 and the fact that $\sigma_g \geq D/p_g$

$$\sup_{\beta \in \mathcal{F}_d} \sum_{j \geq j_0} \sum_{\lambda \in \Lambda} \beta^f_\lambda \beta^p_\lambda$$

$$\leq L_D \sum_{j \geq j_0} 2^{-j(\sigma_d + D/2 - D/p_d)} \|\beta^p\|_{p'_d}$$

$$= L_D \sum_{j \geq j_0} \frac{2^{j(\sigma_g + D/2 - D/p_g)}}{2^{j(\sigma_d + \sigma_g + D - D/p_d - D/p_g)}} 2^{j(D/p'_d - D/p_g)_+} \|\beta^p\|_{p_g}$$

$$\leq L_D \sum_{j \geq j_0} \frac{2^{j(D/p'_d - D/p_g)_+}}{2^{j(\sigma_d + \sigma_g + D/p'_d - D/p_g)}} \sup_{j \geq j_0} 2^{j(\sigma_g + D/2 - D/p_g)} \|\beta^p\|_{p_g}$$

$$\leq 2^{-j_0(\sigma_d + \sigma_g - (D/p_g - D/p'_d)_+)} \|p\|_{p_g q_g}^{\sigma_g} \qquad\qquad (\sigma_g \geq D/p_g)$$

$$\leq c 2^{-j_0(\sigma_d + \sigma_g - (D/p_g - D/p'_d)_+)}$$

$\square$

Using lemmas 21 and 20 we get the following upper bound on the bias and variance of the linear wavelet estimator.

$$c\left( n^{-1/2} + n^{-1/2} 2^{j_0(D/2 - \sigma_d)} + 2^{-j_0(\sigma_g + \sigma_d - D/p_g + D - D/p_d)} \right)$$

which when minimized for $j_0$ gives,

$$2^{j_0} = n^{1/(2\sigma_g + D + 2D/p'_d - 2D/p_g)}$$

which implies an upper bound of

$$\lesssim n^{-1/2} + n^{-\frac{\sigma_g + \sigma_d - D/p_g + D - D/p_d}{2\sigma_g + D + 2D/p'_d - 2D/p_g}}$$

as desired.

## C   Proof of the Lower Bound

In this section we prove our main lower bound i.e. Theorem 4 using Fano's lemma and the Varshamov Gilbert bound as summarized below.

**Lemma 22.** *(Fano's Lemma; Simplified Form of Theorem 2.5 of [53])*

*Fix a family $\mathcal{P}$ of distributions over a sample space $\mathcal{X}$ and fix a pseudo-metric $\rho : \mathcal{P} \times \mathcal{P} \to [0, \infty]$ over $\mathcal{P}$. Suppose there exists a set $T \subseteq \mathcal{P}$ such that there is a $p_0 \in T$ with $p \ll p_0 \ \forall p \in T$ and*

$$s := \inf_{p,p' \in T} \rho(p, p') > 0 \quad , \quad \sup_{p \in T} D_{KL}(p, p_0) \leq \frac{\log |T|}{16},$$

*where $D_{KL} : \mathcal{P} \times \mathcal{P} \to [0, \infty]$ denotes Kullback-Leibler divergence. Then,*

$$\inf_{\widehat{p}} \sup_{p \in \mathcal{P}} \mathbb{E}\left[ \rho(p, \widehat{p}) \right] \geq \frac{s}{16}$$

*where the $\inf$ is taken over all estimators $\widehat{p}$.*

**Lemma 23.** *(Varshamov-Gilbert bound ([53])) Let $\Omega = \{0, 1\}^m$ where $m \geq 8$. Then there exists a subset $\{w^0, \dots, w^M\}$ of $\Omega$ such that $w^0 = (0, \dots, 0)$ and*

$$\omega(w^j, w^k) \geq \frac{m}{8} \quad \forall 0 \leq j, k \leq M$$

*where $M \geq 2^{m/8}$, where $\omega(w^j, w^k) = \sum_{i=1}^m 1_{\{w^j_i \neq w^k_i\}}$ is the Hamming distance.*

*Proof.* (of Theorem 4) We follow the method in Donoho et. al. [14] and separate our proof into "sparse" and "dense" cases. As is standard procedure, for both cases we pick a finite subset of densities from $\mathcal{F}_g$ over which estimation is difficult. Since any function in a Besov space can be defined by its wavelet coefficients we pick a set of densities by an appropriate choice of wavelet coefficients.

Here we also need to pick a subset of functions from $\mathcal{F}_d$ so as to estimate $d_{\mathcal{F}_d}$. Following the method in [47] we pick from $\mathcal{F}_d$, functions that are analogous to the ones we pick from $\mathcal{F}_g$ so that we measure the difference in the densities along the chosen perturbations.

We now fill in the details. We first let $g_0$ be a density function supported on an interval that contains $[-A, A]^D$ such that $\|g_0\|_{\sigma_g p_g q_g} \leq L_G/2$ and $g_0 = c > 0$ on $[-A, A]^D$.

At a particular resolution $j$, we choose $2^{Dj}$ wavelets with disjoint supports; pick $\psi_\lambda = 2^{Dj/2}\psi_{\epsilon_1}(2^{Dj}x - k)$ indexed by $\lambda = 2^{-j}k + 2^{-(j+1)}\epsilon_1$ s.t. $k \in K_j$ where

$$K_j = \{-(2^j - 1)A + 2lA, l = 0, \dots, (2^j - 1)\}^D$$

and $\epsilon_1 = (1, 0, \dots, 0)$ (i.e. we pick the first wavelet). Note here that if $\lambda \neq \lambda'$ then $\psi_\lambda$ and $\psi_{\lambda'}$ have disjoint support.

We now describe our choice of densities based on the set of coefficients $\zeta \subseteq \{\tau \in \mathbb{Z}^{|K_j|} : |\tau_\lambda| \leq 1\}$ i.e.

$$\Omega_g := \{g_0 + c_g \sum_\lambda \tau_\lambda \psi_\lambda : \tau \in \zeta, \lambda = 2^{-j}k + 2^{-j-1}\epsilon_1, k \in K_j\}.$$

If we pick $c_g$ to be small enough, every $p$ in $\Omega_g$ is a density function and is lower bounded on $[-A, A]^D$. Specifically if $c_g$ s.t.

$$c_g \leq \frac{c}{2\|\psi\|_\infty}2^{-Dj/2}$$

then $\int g_0 + c_g \sum_\lambda \tau_\lambda \psi_\lambda = 1$ (since $\int \psi_\lambda = 0$) and,

$$\|g_0 - p\|_\infty = c_g 2^{Dj/2}\|\psi\|_\infty \leq c/2$$

so that $p$ is lower bounded on the domain of $\psi_\lambda$ by $c/2$ for every $\lambda$. This also implies that $p$ is always positive.

Now the following lemma states that if you have a small perturbation of a density s.t. the density is lower bounded on the support of the perturbation then the KL divergence between the perturbed and the original density is upper bounded by the $L^2$ norm of the perturbation.

**Lemma 24.** *Let $g = g_0 + h$, $g_0$ be density functions such that $h \leq g_0$. If $S = supp(h) \subseteq supp(g)$ and $c \leq g$ on $S$, where $c$ is a constant. Then*

$$D_{KL}(g^n, g_0^n) \leq cn\|g_0 - g\|_{L^2}^2$$

*Proof.* Since $g \leq 2g_0$ we have,

$$\frac{g_0 - g}{g} \geq -\frac{1}{2}$$

so using the fact that $-\log(1 + x) \leq x^2 - x$ for all $x \geq -1/2$ we get

$$\begin{aligned}
D_{KL}(g^n, g_0^n) &= nD_{KL}(g, g_0)\\
&= n\int_S g(x)\log\frac{g(x)}{g_0(x)}\,dx\\
&= -n\int_S g(x)\log\left(1 + \frac{g_0(x) - g(x)}{g(x)}\right)dx\\
&\leq n\int_S g(x)\left(\left(\frac{g_0(x) - g(x)}{g(x)}\right)^2 - \frac{g_0(x) - g(x)}{g(x)}\right)dx\\
&= n\int_S \frac{(g_0(x) - g(x))^2}{g(x)}\,dx
\end{aligned}$$

which, since $g \geq c$ on $S$, is smaller than $cn\int_S (g_0(x) - g(x))^2$ as desired. $\qquad\square$

Using this fact we conclude that for any $p_\tau \in \Omega_g$,

$$KL(p_\tau, g_0) \leq nc_g^2 c \left\| \sum_\lambda \tau_\lambda \psi_\lambda \right\|_{L^2}^2 = cnc_g^2 \|\tau\|_2^2$$

Following the technique in [48] we also pick an analogous set of functions that live in $\mathcal{F}_d$ so that we can lower bound $d_{\mathcal{F}_D}$. In particular let

$$\Omega_d := \{c_d \sum_\lambda \tau_\lambda \psi_\lambda : \tau \in \zeta, \lambda = 2^{-j}k + 2^{-j-1}\epsilon_1, k \in K_j\}$$

It now, only remains to choose appropriate sets $\zeta$ for the wavelet coefficients in each of the sparse and dense cases. In the remainder let $c$ be a constant not necessarily the same.

**Sparse or low-smoothness case:**

For the sparse/lower smoothness case we choose worst case densities to be perturbations along only a specific scaling of the wavelet at a time. In particular, let

$$\zeta = \{\tau : \tau_\lambda = 1, \tau_{\lambda'} = 0, \lambda' \neq \lambda = 2^{-j}k + 2^{-(j+1)}\epsilon_1, k \in K_j\}$$

We know from above that for any $c_g \leq c2^{-Dj/2}$, every $p \in \Omega_g$ is a density such that $D_{KL}(p^n, g_0^n) \leq cnc_g^2 \|\tau\|_2$. Now, we need

$$\|g_0 + c_g\psi_\lambda\|_{p_g q_g}^{\sigma_g} \leq \|g_0\|_{p_g q_g}^{\sigma_g} + 2^{j(\sigma_g + D/2 - D/p_g)}c_g \leq L_g$$

so that $\Omega_g \subseteq \mathcal{F}_g$. Since $\sigma_g \geq D/p_g$ the choice of $c_g = c2^{-j(\sigma_g + D/2 - D/p_g)}$ suffices. Similarly, $c_d = L_d 2^{-j(\sigma_d + D/2 - D/p_d)}$ implies $\Omega_d \subseteq \mathcal{F}_d$.

Then we pick $j$ large enough such that the KL divergence between any $p_\tau$ and $g_0$ is small. This enables us to apply Fano's lemma from above and get a lower bound.

So we need $cnc_g^2 \leq \frac{\log|\zeta|}{16} = \frac{\log|K_j|}{16}$ i.e.

$$n \leq cj/c_g^2 \iff n \leq 2^{2j(\sigma_g + D/2 - D/p_g)}j$$

for the KL divergence to be small. Given such a $j$ we have,

$$d_{\mathcal{F}_d}(p_\lambda, p_{\lambda'}) \geq \sup_{f \in \Omega_d} \left| \int c_g(f(x)(\psi_\lambda - \psi_{\lambda'})dx \right| = \|\psi\|_{L^2}^2 c_g c_d$$

(since, $\|\psi_\lambda\|_{L^2}^2 = \|\psi\|_{L^2}^2$). So, if $2^j = (n/\log n)^{\frac{1}{2\sigma_g + D - 2D/p_g}}$ we have,

$$M(\mathcal{F}_g, \mathcal{F}_d) \gtrsim \left( \frac{\log n}{n} \right)^{\frac{\sigma_g + \sigma_d + D - D/p_g - D/p_d}{2\sigma_g + D - 2D/p_g}}$$

**Dense or higher smoothness case:**

In the dense case, we choose our set of densities by perturbing $g_0$ along every scaling of the wavelet simultaneously i.e. let

$$\zeta = \{\tau : \tau_\lambda \in \{-1, +1\}\}$$

Now, we need

$$\left\| g_0 + c_g \sum_\lambda \tau_\lambda \psi_\lambda \right\|_{p_g q_g}^{\sigma_g} \leq \|g_0\|_{p_g q_g}^{\sigma_g} + 2^{j(\sigma_g + D/2)}c_g \leq L_g$$

so that $\Omega_g \subseteq \mathcal{F}_g$. The choice of $c_g = c2^{-j(\sigma_g + D/2)}$ suffices. Similarly, $c_d = L_d 2^{-j(\sigma_d + D/2)}$ implies $\Omega_d \subseteq \mathcal{F}_d$.

Now the Varshamov-Gilbert bound from above implies we can pick a subset of $\Omega_G$ with size at least $2^{|K_j|/8}$ such that $\omega(\tau_\lambda, \tau_{k'}) \geq |K_j|/8$ which gives,

$$d_{\mathcal{F}_d}(p_\lambda, p_{\lambda'}) = \sup_{f \in \Omega_d} \left| \int c_g(f(x)(\psi_\lambda - \psi_{\lambda'})dx \right|$$

$$= c_g c_d \omega(\tau_\lambda, \tau_{\lambda'}) \geq c_g c_d \frac{2^{Dj}}{4}$$

We pick $j$ large enough such that the KL divergence between any $p_\tau$ and $g_0$ is small. This enables us to apply Fano's lemma from above and get a lower bound. In particular we need, for any $p_\tau \in \Omega_g$, $D_{KL}(p_\tau^n, g_0^n) \leq cnc_g^2 \|\tau\|_2 = cnc_g^2 |K_j|$ to be at most $\frac{\log|\zeta|}{16} = \frac{|K_j|}{16}$ which is equivalent to $n \leq 2^{j(2\sigma_g + D)}$. Then by Fano's lemma the lower bound in the dense case is

$$n^{-\frac{\sigma_g + \sigma_d}{2\sigma_g + D}}$$

We combine the above two cases to get the following lower bound on the rate

$$\gtrsim \max \left( n^{-\frac{\sigma_g + \sigma_d}{2\sigma_g + D}}, n^{-\frac{\sigma_g + \sigma_d + D - D/p_g - D/p_d}{2\sigma_g + D - 2D/p_g}} \right)$$

□

## D   Proof of the Upper Bound

We use the wavelet thresholding estimate as introduced in [14] to get an upper bound on our minimax rate.

*Proof.* (of theorem 5) We first upper bound our error by three terms namely, the stochastic error, the bias and the non-linear terms. The stochastic error is bounded above as usual by the above moment bound. The bias is bounded above by virtue of our density belonging to the besov space $B_{p_g, q_g}^{\sigma_g}$. The non-linear terms are more delicate. We follow the procedure in [14] and split them into four groups the first two of which are shown to be negligible as the probability of large deviations falls exponentially rapidly from Bernstein's inequality above. We simplify the upper bounds on the other two terms considerably by paying a penalty on the rate by the factor that is logarithmic in the sample size. We now fill in the details of the proof.

We first let our discriminator and generator classes be

$$\mathcal{F}_d = \{f : \|f\|_{p_d, q_d}^{\sigma_d} \leq L_d\}$$
$$\mathcal{F}_g = \{p : \|p\|_{p_g, q_g}^{\sigma_g} \leq L_g\} \cap \mathcal{P}$$
$$\mathcal{P} = \{p : p \geq 0, \|p\|_{L^1} = 1, \operatorname{supp}(p) \subseteq [-T, T]\}$$

Given $X_1, \ldots, X_n$ be IID with density $p \in \mathcal{F}_g$ and the thresholded wavelet estimator of $p$ i.e.

$$p = \sum_{k \in \mathbb{Z}} \alpha_k^p \phi_k + \sum_{j \geq 0} \sum_{\lambda \in \Lambda_j} \beta_\lambda^p \psi_\lambda$$

$$\widehat{p}_n = \sum_{k \in \mathbb{Z}} \widehat{\alpha}_k \phi_k + \sum_{j=0}^{j_0} \sum_{\lambda \in \Lambda_j} \widehat{\beta}_\lambda \psi_\lambda + \sum_{j=j_0}^{j_1} \sum_{\lambda \in \Lambda_j} \widetilde{\beta}_\lambda \psi_\lambda$$

where

$$\alpha_k^p = \mathop{\mathbb{E}}_{X \sim p} [\phi_k(X)] \qquad \widehat{\alpha}_k = \frac{1}{n} \sum_{i=1}^n \phi_k(X_i)$$
$$\beta_\lambda^p = \mathop{\mathbb{E}}_{X \sim p} [\psi_\lambda(X)] \qquad \widehat{\beta}_\lambda = \frac{1}{n} \sum_{i=1}^n \psi_\lambda(X_i) \qquad \widetilde{\beta}_\lambda = \widehat{\beta}_\lambda \mathbf{1}_{\{\widehat{\beta}_\lambda > t\}}$$

with $t = K\sqrt{j/n}$, where $K$ is a constant to be specified later, and

$$2^{j_0} = n^{\frac{1}{2\sigma_g + D}}$$

$$2^{j_1} = n^{\frac{1}{2\sigma_g + D - 2D/p_g}}$$

we can upper bound the error as,

$$d_{\mathcal{F}_d}(p, \widehat{p}_n) \leq \sup_{f \in \mathcal{F}_d} \sum_{k \in \mathbb{Z}} \alpha_k^f (\alpha_k^p - \widehat{\alpha}_k) \qquad + \sup_{f \in \mathcal{F}_d} \sum_{j=0}^{j_0} \sum_{\lambda \in \Lambda_j} \beta_\lambda^f \left( \beta_\lambda^p - \widehat{\beta}_\lambda \right)$$

$$+ \sup_{f \in \mathcal{F}_d} \sum_{j \geq j_0}^{j_1} \sum_{\lambda \in \Lambda_j} \beta_\lambda^f \left( \beta_\lambda^p - \widetilde{\beta}_\lambda \right) \qquad + \sup_{f \in \mathcal{F}_d} \sum_{j \geq j_1} \sum_{\lambda \in \Lambda_j} \beta_\lambda^f \beta_\lambda^p$$

where the first three terms constitute the stochastic error (the non-linear terms or thresholded terms are also called 'detail' terms [14]) and the last term is the bias. In particular:

1. The first term in our upper bound of the risk is the stochastic error or the variance of a linear wavelet estimator with $j_0$ terms. Note that since $\sigma_g \geq D/p_g$ $p \in \mathcal{F}_g$ implies by lemma 17 that $\|p\|_\infty < \infty$. Then by substitution

$$\mathbb{E}_p |\psi_\lambda(X)|^{p_d'} \leq 2^{-Dj(p_d'/2 - 1)}$$

Therefore by lemma 20 we have an upper bound here of

$$cn^{-1/2}(2^{j_0(D/2 - \sigma_d)} + 1) \lesssim n^{-\frac{\sigma_g + \sigma_d}{2\sigma_g + D}} + n^{-1/2}$$

2. The third term is the bias of a linear wavelet estimator with $j_1$ terms which by lemma 21 for $p_d' \geq p_g$ is bounded above by

$$c2^{-j_1(\sigma_d + \sigma_g - D/p_g + D/p_d')} \lesssim n^{-\frac{\sigma_g + \sigma_d + D - D/p_g - D/p_d}{2\sigma_g + D - 2D/p_g}}$$

3. For the second term we have, by lemmas 13 and 16

$$\mathbb{E} \sup_{f \in \mathcal{F}_d} \sum_{j \geq j_0}^{j_1} \sum_{\lambda \in \Lambda} \beta_\lambda^f \left( \beta_\lambda^p - \widetilde{\beta}_\lambda \right) \leq L_D \sum_{j=j_0}^{j_1} 2^{-j(\sigma_d + D/2 - D/p_d)} \left( \mathbb{E} \sum_{\lambda \in \Lambda_j} |\beta_\lambda^p - \widetilde{\beta}_\lambda|^{p_d'} 1_A \right)^{1/p_d'}$$

$$\leq L_D \sum_{j=j_0}^{j_1} 2^{-j(\sigma_d + D/2 - D/p_d)} \left( \sum_{\lambda \in \Lambda_j} \mathbb{E} |\beta_\lambda^p - \widetilde{\beta}_\lambda|^{p_d'} 1_A \right)^{1/p_d'}$$

where we are only summing over finitely many terms. The set $A$ is given by the following cases:

(For the upper bounds of the first two cases we have chosen $\gamma$ (which in turn determines the value of $K$) to be large enough so that the exponent of $2^j$ is negative and thus we can upper bound the geometric series by a constant multiple of the first term.)

(a) Let $A$ be the set of $k$ s.t. $\widehat{\beta}_\lambda > t$ and $\beta_\lambda^p < t/2$ and $r \geq 1/p_d'$ then

$$L_D \sum_{j=j_0}^{j_1} 2^{-j(\sigma_d + D/2 - D/p_d)} \left( \sum_{\lambda \in \Lambda_j} \mathbb{E} |\beta_\lambda^p - \widetilde{\beta}_\lambda|^{p_d'} 1_A \right)^{1/p_d'}$$

$$\leq L_D \sum_{j=j_0}^{j_1} 2^{-j(\sigma_d + D/2 - D/p_d)} \left( \sum_{\lambda \in \Lambda_j} (\mathbb{E} |\beta_\lambda^p - \widetilde{\beta}_\lambda|^{p_d' r})^{1/r} \Pr(A)^{1/r'} \right)^{1/p_d'}$$

Using the large deviation and moment bound

$$\Pr(A) \le \Pr\left(|\widehat{\beta}_\lambda - \beta_\lambda^p| \ge t/2\right) \le c 2^{-\gamma j}$$

we get,

$$\le c \sum_{j=j_0}^{j_1} 2^{-j(\sigma_d + D/2 - D/p_d)} \left(2^{Dj} n^{-p'_d/2} 2^{-j\gamma/r'}\right)^{1/p'_d}$$

$$\le c \sum_{j=j_0}^{j_1} 2^{-j(\sigma_d + D/2 - D/p_d - D/p'_d)} n^{-1/2} 2^{-\gamma j/p'_d r'}$$

$$\le c n^{-1/2} \sum_{j=j_0}^{j_1} 2^{-j(\sigma_d - D/2 + \gamma/p'_d r')}$$

$$\le c n^{-1/2} 2^{-j_0(\sigma_d - D/2 + \gamma/p'_d r')}$$

$$\lesssim n^{-\frac{\sigma_g + \sigma_d + \gamma/p'_d r'}{2\sigma_g + D}},$$

which is negligible compared to the linear term.

(b) Let $B$ be the set of $k$ s.t. $\widehat{\beta}_\lambda < t$ and $\beta_\lambda^p > 2t$ then same as above

$$\mathbb{E} \sup_{f \in \mathcal{F}_d} \sum_{j=j_0}^{j_1} \sum_{\lambda \in \Lambda_j} \beta_\lambda^f \beta_\lambda^p 1_B \le L_D \sum_{j=j_0}^{j_1} 2^{-j(\sigma_d + D/2 - D/p_d)} \left\|\beta_\lambda^p\right\|_{p'_d} (\Pr(B))^{1/p'_d}$$

$$\le L_D \sum_{j=j_0}^{j_1} 2^{-j(\sigma_d + D/2 - D/p_d)} \left\|\beta_\lambda^p\right\|_{p'_d} 2^{-\gamma j/p'_d}$$

$$\le L_D \sum_{j=j_0}^{j_1} 2^{-j(\sigma_d + \sigma'_g + \gamma/p'_d)} \sup_{j_0 \le j \le j_1} 2^{j(\sigma'_g + D/2 - D/p'_d)} \left\|\beta_\lambda\right\|_{p'_d}$$

$$\le L_D L_G \sum_{j=j_0}^{j_1} 2^{-j(\sigma_d + \sigma'_g + \gamma/p'_d)}$$

$$\le L_D L_G C 2^{-j_0(\sigma_d + \sigma'_g + \gamma/p'_d)}$$

$$\lesssim n^{-\frac{\sigma_d + \sigma'_g + \gamma}{2\sigma_g + D}}$$

which is negligible compared to the bias term.

(c) Let $C$ be the set of $k$ s.t. $\widehat{\beta}_\lambda > t$ and $\beta_\lambda^p > t/2$ then:

$$\mathbb{E} \sup_{f \in \mathcal{F}_d} \sum_{j=j_0}^{j_1} \sum_{\lambda \in \Lambda_j} \beta_\lambda^f \left( \beta_\lambda^p - \widetilde{\beta}_\lambda \right) 1_C$$

$$\leq L_D \sum_{j=j_0}^{j_1} 2^{-j(\sigma_d + D/2 - D/p_d)} \left( \sum_{k \in C} \mathbb{E} \, |\beta_\lambda^p - \widetilde{\beta}_\lambda|^{p_d'} \right)^{1/p_d'}$$

$$\leq L_D \sum_{j=j_0}^{j_1} C n^{-1/2} 2^{-j(\sigma_d + D/2 - D/p_d)} \left( \sum_{k \in C} \left( \frac{2\beta_\lambda^p \sqrt{n/j}}{K} \right)^{p_g} \right)^{1/p_d'}$$

$$\leq L_D \sum_{j=j_0}^{j_1} C n^{-1/2} (\sqrt{n/j})^{p_g/p_d'} 2^{-j(\sigma_d + D/2 - D/p_d)} \|\beta^p\|_{p_g}^{p_g/p_d'}$$

$$\leq L_D \sum_{j=j_0}^{j_1} C n^{-1/2} (\sqrt{n/j})^{p_g/p_d'} 2^{-j(\sigma_d + D/2 - D/p_d)} 2^{-j(\sigma_g + D/2 - D/p_g)p_g/p_d'}$$

$$\sup_{j_0 \leq j \leq j_1} \|\beta\|_{p_g} 2^{j(\sigma_g + D/2 - D/p_g)}$$

$$\leq C L_D L_G n^{1/2(p_g/p_d' - 1)} \sum_{j=j_0}^{j_1} 2^{-j((\sigma_g + D/2)p_g/p_d' + \sigma_d - D/2)} j^{-p_g/2p_d'}$$

$$\leq C L_D L_G n^{1/2(p_g/p_d' - 1)} 2^{-j_m((\sigma_g + D/2)p_g/p_d' + \sigma_d - D/2)}$$

where

$$j_m = \begin{cases} j_0 & (2\sigma_g + D)p_g \geq (D - 2\sigma_d)p_d' \\ j_1 & (2\sigma_g + D)p_g \leq (D - 2\sigma_d)p_d' \end{cases}$$

In the first case we have an upper bound of

$$\lesssim n^{-\frac{\sigma_g + \sigma_d}{2\sigma_g + D}}$$

and in the second case we have an upper bound of

$$\lesssim n^{-\frac{\sigma_g + \sigma_d + D - D/p_d - D/p_g}{2\sigma_g + D - 2D/p_g}}$$

(d) Let $E$ be the set of $k$ s.t. $\widehat{\beta}_\lambda < t$ and $\beta_\lambda^p < 2t$ then:

$$\mathbb{E} \sup_{f \in \mathcal{F}_d} \sum_{j=j_0}^{j_1} \sum_{\lambda \in \Lambda_j} \beta_\lambda^f \beta_\lambda^p 1_D$$

$$\leq L_D \sum_{j=j_0}^{j_1} 2^{-j(\sigma_d + D/2 - D/p_d)} \left( \sum_{\lambda \in \Lambda_j} |\beta_\lambda^p|^{p_d'} \right)^{1/p_d'}$$

$$\leq L_D \sum_{j=j_0}^{j_1} 2^{-j(\sigma_d + D/2 - D/p_d)} \left( \sum_{\lambda \in \Lambda_j} |\beta_\lambda^p|^{p_g} (2t)^{p_d' - p_g} \right)^{1/p_d'} \qquad p_d' \geq p_g$$

$$= L_D \sum_{j=j_0}^{j_1} 2^{-j(\sigma_d + D/2 - D/p_d)} (2t)^{1 - p_g/p_d'} \|\beta\|^{p_g/p_d'}$$

$$\leq L_D \sum_{j=j_0}^{j_1} 2^{-j(\sigma_d + D/2 - D/p_d)} (2\sqrt{j/n})^{1 - p_g/p_d'} 2^{-j(\sigma_g + D/2 - D/p_g)p_g/p_d'} L_g$$

$$\leq c \sqrt{j_1} n^{1/2(p_g/p_d' - 1)} \sum_{j=j_0}^{j_1} 2^{-j((\sigma_g + D/2)p_g/p_d' + \sigma_d - D/2)} j^{-p_g/2p_d'}$$

$$\lesssim \left( n^{-\frac{\sigma_g + \sigma_d}{2\sigma_g + D}} + n^{-\frac{\sigma_g + \sigma_d + D - D/p_d - D/p_g}{2\sigma_g + D - 2D/p_g}} \right) \sqrt{\log n}$$

$\square$

# E  Proof of Theorem 7

**Lower Bound**

*Proof.* Just as in the proof of the lower bound above we let $j \geq 0$ and

$$\Omega_g := \{g_0 \pm c_g \psi_\lambda : \lambda = 2^{-j}k + 2^{-j-1}\epsilon_1, k \in K_j\}$$

where $\epsilon_1 = (1, 0, \ldots, 0)$. Here we let $g_0 = 2^{Dj}c$ on at least $[-A, A]^D$ and

$$c_g = \min\left(\frac{c}{2\|\psi\|_\infty}2^{-Dj/2}, \frac{L_g}{2}2^{-j(\sigma_g + D/2 - D/p_g)}\right)$$

such that $\Omega_g \subseteq \mathcal{F}_g$. We also let

$$\Omega_d := \{c_d \sum_\lambda \tau_\lambda \psi_\lambda : \lambda = 2^{-j}k + 2^{-j-1}\epsilon_1, k \in K_j, \|\tau\| \leq L_d\}$$

s.t.

$$c_d \leq L_d 2^{-j(\sigma_d + D/2 - 1/p_d)}$$

i.e. $\Omega_d \subseteq \mathcal{F}_d$.

Then for any linear estimate $\widehat{P}$ with $\widehat{\alpha}_\lambda = \int \psi_\lambda(x)d\widehat{P}(x)$,

$$\sup_{P \in \mathcal{F}_g} \mathbb{E}_P \sup_{f \in \mathcal{F}_d} \left|\int f(x)(dP(x) - d\widehat{P}(x))\right|$$

$$\geq \sup_{p \in \Omega_g} \mathbb{E}_P \sup_{f \in \Omega_d} \left|\int f(x)(p(x)dx - d\widehat{P}(x))\right|$$

$$= \sup_{\lambda:k \in K_j} \frac{c_d}{2} \mathbb{E}_{g_0 + c_g\psi_\lambda} \left(\sup_{\tau:\|\tau\|_{p_d} \leq L_d} \sum_{\lambda' \neq \lambda} |\tau_{\lambda'}\widehat{\alpha}_{\lambda'}| + |\tau_\lambda||c_g - \widehat{\alpha}_\lambda|\right)$$

$$+ \mathbb{E}_{g_0 - c_g\psi_\lambda} \left(\sup_{\tau:\|\tau\|_{p_d} \leq L_d} \sum_{\lambda' \neq \lambda} |\tau_{\lambda'}\widehat{\alpha}_{\lambda'}| + |\tau_\lambda||c_g - \widehat{\alpha}_\lambda|\right)$$

$$\geq \sup_{\lambda:k \in K_j} \frac{c_d}{2}$$

$$\sup_{\tau:\|\tau\|_{p_d} \leq L_d} \left(\sum_{\lambda' \neq \lambda} \mathbb{E}_{g_0 + c_g\psi_\lambda} |\tau_{\lambda'}||\widehat{\alpha}_{\lambda'}| + \mathbb{E}_{g_0 - c_g\psi_\lambda} |\tau_{\lambda'}||\widehat{\alpha}_{\lambda'}| + \mathbb{E}_{g_0 + c_g\psi_\lambda} |\tau_\lambda||c_g - \widehat{\alpha}_\lambda| + \mathbb{E}_{g_0 - c_g\psi_\lambda} |\tau_\lambda||c_g - \widehat{\alpha}_\lambda|\right)$$

$$= \sup_{\lambda:k \in K_j} \frac{c_d}{2}$$

$$\left(\sum_{\lambda' \neq \lambda} (\mathbb{E}_{g_0 + c_g\psi_\lambda} |\widehat{\alpha}_{\lambda'}|)^{p'_d} + (\mathbb{E}_{g_0 - c_g\psi_\lambda} |\widehat{\alpha}_{\lambda'}|)^{p'_d} + (\mathbb{E}_{g_0 + c_g\psi_\lambda} |c_g - \widehat{\alpha}_\lambda|)^{p'_d} + (\mathbb{E}_{g_0 - c_g\psi_\lambda} |c_g - \widehat{\alpha}_\lambda|)^{p'_d}\right)^{1/p'_d}$$

$$\geq c_d \left(\frac{1}{2^{Dj}} \sum_{\lambda' \neq \lambda} (\mathbb{E}_{g_0 + c_g\psi_\lambda} |\widehat{\alpha}_{\lambda'}|)^{p'_d} + (\mathbb{E}_{g_0 - c_g\psi_\lambda} |\widehat{\alpha}_{\lambda'}|)^{p'_d} + (\mathbb{E}_{g_0 + c_g\psi_\lambda} |c_g - \widehat{\alpha}_\lambda|)^{p'_d} + (\mathbb{E}_{g_0 - c_g\psi_\lambda} |c_g - \widehat{\alpha}_\lambda|)^{p'_d}\right)^{1/p'_d}$$

Now the expression inside the brackets is bounded below in [14] appendix A.3 by $n^{-1/2}2^{jD/p'_d}$ where $2^j = n^{\frac{1}{2\sigma_g - 2D/p_g + 2D/p'_d + D}}$ which implies a lower bound in our case of

$$c2^{-j(\sigma_d + D/2 - D/p_d)}n^{-1/2}2^{Dj/p'_d}$$

$$= c2^{j(D/2 - \sigma_d)}n^{-1/2}$$

which gives us a lower bound of

$$\gtrsim n^{-\frac{\sigma_d+\sigma_g-D/p_g+D/p_d'}{2\sigma_g-2D/p_g+2D/p_d'+D}}$$

as desired.

$\square$

# F  Proof of Theorem 9

Here, we prove the following theorem, which upper bounds the risk of an appropriately constructed GAN for learning Besov distributions:

**Theorem 25** (Convergence Rate of a Well-Optimized GAN). *Fix a Besov density class $B_{p_g,q_g}^{\sigma_g}$ with $\sigma_g > D/p_g$ and discriminator class $B_{p_d,q_d}^{\sigma_d}$ with $\sigma_d > D/p_d$. Then, for any desired approximation error $\epsilon > 0$, one can construct a GAN $\widehat{p}$ of the form (9) (with $\widetilde{p}_n$) with discriminator network $N_d \in \Phi(H_d, W_d, S_d, B_d)$ and generator network $N_g \in \Phi(H_g, W_g, S_g, B_g)$, s.t. for all $p \in B_{p_g,q_g}^{\sigma_g}$*

$$\mathbb{E}\left[d_{B_{p_d,q_d}^{\sigma_d}}\left(\widehat{p}, p\right)\right] \lesssim \epsilon + \mathbb{E}\, d_{B_{p_d,q_d}^{\sigma_d}}\left(\widetilde{p}_n, p\right)$$

*where $H_d$, $H_g$ grow logarithmically with $1/\epsilon$, $W_d, S_d, B_d, W_g, S_g, B_g$ grow polynomially with $1/\epsilon$ and $C > 0$ is a constant that depends only on $B_{p_d,q_d}^{\sigma_d}$ and $B_{p_g,q_g}^{\sigma_g}$.*

Our statistical guarantees rely on a recent construction, by Suzuki [51], of a fully-connected ReLU network that approximates Besov functions. Specifically, we leverage the following result:

**Lemma 26** (Proposition 1 of Suzuki [51]). *Suppose that $p, q, r \in (0, \infty]$ and $\sigma > \delta := D(1/p - 1/r)_+$ and let $\nu = (\sigma - \delta)/(2\delta)$. Then, for sufficiently small $\epsilon \in (0, 1)$, there exists a constant $C > 0$, depending only on $D, p, q, r, \sigma$, such that, for some*

$$H \le C \log(1/\epsilon), \quad W \le C\epsilon^{-D/\sigma}, \quad S \le C\epsilon^{-D/\sigma} \log(1/\epsilon), \quad B \le C\epsilon^{-(D/\nu+1)(1\vee(D/p-\sigma)_+)/\sigma},$$

*$\Phi(H, W, S, B) \subseteq B_{p,q}^{\sigma}(1)$ and $\Phi(H, W, S, B)$ approximates $B_{p,q}^{\sigma}(1)$ to accuracy $\epsilon$ in $L^r$; i.e.,*
$$\sup_{f \in B_{p,q}^{\sigma}(1)} \inf_{\widetilde{f} \in \Phi(H,W,S,B)} \|f - \widetilde{f}\|_{L^r} \le C\epsilon.$$

*Proof.* Liang [28, Inequality 2.2] showed that we can decompose the error, for densities $\widehat{p}$, $p$,

$$d_{\mathcal{F}_d}\left(\widehat{p}, p\right) \le \inf_{q \in \Phi(H_g, W_g, S_g, B_g)} d_{\mathcal{F}_d}\left(p, q\right)$$
$$+ 2 \sup_{f \in \mathcal{F}_d} \inf_{g \in \Phi(H_d, W_d, S_d, B_d)} \|f - g\|_\infty$$
$$+ d_{\Phi(H_d, W_d, S_d, B_d)}\left(p, \widetilde{p}_n\right) + d_{\mathcal{F}_d}\left(p, \widetilde{p}_n\right),$$

where the 3 summands above correspond respectively the error of approximating $\mathcal{F}_g$ by $\Phi(L_g, W_g, S_g, B_g)$ (generator approximation error), the error of approximating $\mathcal{F}_d$ by $\Phi(L_d, W_d, S_d, B_d)$ (discriminator approximation error), and statistical error.

To bound the first term, note also that, since we assumed $\sigma_d > D/p_d$, we have the embedding $B_{p_d,q_d}^{\sigma_d} \subseteq L^\infty$, and, in particular, $M := \sup_{f \in B_{p_d,q_d}^{\sigma_d}} \|f\|_{L^\infty} < \infty$. Thus, by Hölder's inequality, the assumption that densities in $\mathcal{P}$ are supported only on $[-T, T]$, and Lemma 26 (with $r = \infty$),

$$\inf_{q \in \mathcal{F}_g} d_{\mathcal{F}_d}\left(p, q\right) \le \inf_{q \in \mathcal{F}_g}\left(p, q\right) \sup_{f \in \mathcal{F}_D} \|f\|_{L^1([-T,T])} \|p - q\|_{L^\infty} \le 2MT\epsilon.$$

To bound the second term, simply observe that, by Lemma 26 (with $r = \infty$),

$$\sup_{f \in \mathcal{F}_d} \inf_{g \in \phi(L_g, W_g, S_g, B_g)} \|f - g\|_\infty \le \epsilon.$$

Since, by Lemma 26, $\Phi(L_d, W_d, S_d, B_d) \subseteq B_{p_d,q_d}^{\sigma_d}$, the last term is immediately bounded (in expectation) by $d_{\mathcal{F}_d}(\widetilde{p}_n, p)$. Combining the bounds on these three terms gives

$$d_{\mathcal{F}_d}\left(\widehat{p}, p\right) \le 2(MT + 1)\epsilon + 2d_{\mathcal{F}_d}(\widetilde{p}_n, p).$$

$\square$

## Footnotes

[3]We assume a good optimization algorithm for computing (1), although this is also an active area of research.

[4] As in these previous works, we assume implicitly that the optimum (9) can be computed; this complex saddle-point problem is itself the subject of a related but distinct and highly active area of work [40, 3, 30, 20].