[Reviews · NeurIPS 2019]

Reviewer 1



The paper is of interest firstly by its technical results. Non parametric estimation with adversarial losses as well as Gans are clearly the focus of numerous current ML research and in my opinion this work is a significant contribution on these subjects. Their rates unify/improve a lot of existing results of the literature and cover a wide variety of settings, thanks to the generality of Besov spaces. The paper is also very well written, clearly introducing to the reader the substantial mathematical content needed on wavelets/Besov spaces and discussing the results in a honest way.Then it provides very interesting interpretation of the results as well as an advanced discussion on the different regimes of smoothness. Generally this paper provides a deeper understanding on this problem. I really appreciated the investment of the authors in the writing and I enjoyed reading this paper. The proofs are also detailed and clearly written. They use classic tools from statistics (Fano's lemma, concentration bounds, wavelet estimators) and look correct - I did not find any flaw.

Reviewer 2



My comments mainly lie in the following perspective: 1. About \epsilon in Theorem 9. I think it may depend on the parameters of Besov space. As the design of both discriminator network and generator network depends on \epsilon, does this imply that the result is not adaptive, i.e., in order the achieve the minimax rate, we need to know \sigma_g, p_g, q_g? 2. Could the authors briefly summarize technical contributions of the paper? It seems that Theorem 9 mainly depends on the previous study of approximation ability of fully-connected ReLU networks. 3. GANs achieves minimax convergence rate over the Besov space, while the same rate can also be achieved by wavelet thresholding. This implies that if we consider Besov space and minimax rate, GANs cannot outperform wavelet thresholding. In order to demonstrate the superiority of GANs, especially in terms of image analysis, is it possible to study some more restrictive function classes other than the Besov space? This comment may be beyond the scope of the paper, but I do think it closely related to the study of statistical properties of GANs. ------------------------------------------------------------- Thank you for the response, which clarifies the adaptivity and theoretical contributions. My score remains the same.

Reviewer 3



- This paper is poorly written and is hard to follow. Its structure and organization could be substantially improved. The notation is unclear, and the terminology is not defined. For example, see lines 45-48. The formal problem statement (section 2.2) is vague, as well. - Unclear novelty. Many technical terms are used without any context; they are not explained and, further, it is not clear how those concepts help the claims made in the paper. For example, in lines 125-156, authors say: "We end this section with a brief survey of known results for estimating distributions under specific Besov IPM losses, noting that our results (Equations (3) and (4) below) generalize all these rates." Authors then spend the rest of that section listing known results about L_p distances, Wasserstein distances, KS and Sobolov distance; however, authors never explain how exactly their results generalize all these known results. The reviewer has an impression as if several parts of the paper were taken directly from a textbook without a proper introduction of terminology. - Unclear relevance to the machine learning community. I understand that this paper describes theoretical work, and because of that, it does not include empirical results and benchmarks. Nevertheless, it is unclear what is a direct relevance of the new results for GANs and nonparametric density estimation. Authors provide a discussion of results in section 5. However, it is unclear and confusing. Again, the notation is not defined; for example, see lines 271-283. -----Update----- The rebuttal addressed many of my concerns, in particular, those related to the terminology used in the paper, and relevance of new results for GANs and nonparametric density estimation. I highly encourage authors to include those additional explanations in the paper and to remove alternative notations (or define in appendix), which are not needed to understand the paper. The author response has convinced me to increase my initial score for this submission.

[Author Response · NeurIPS 2019]

We thank all the reviewers for their comments and suggestions.

**Reviewer #1:** Thank you for the positive review. We will make the corrections you have pointed out.

**Reviewer #2:** We will add discussion of these points to clarify the theoretical contributions of the paper:

1. Yes, Theorem 9 is non-adaptive (i.e., requires knowing $\sigma_g$ to tune the wavelet threshold and generator network
size). In practice, GANs need extensive tuning to perform well, so constructing an "adaptive GAN" could be
useful, but it is not clear to us how to do so. For now, we leave this as future work.
2. Yes, Theorem 9 relies heavily on Theorem 5 and prior work on the approximation ability of fully-connected
ReLU networks. Hence, our main technical contributions are in Theorems 4, 5, and 7. While proving Theorem
9 is straightforward given these results, we nevertheless feel that it is not obvious (and is worth explicitly
sharing with the NeurIPS community) that these results have implications for GANs.
3. Yes, this is an important issue. We provide an example of an acknowledged challenge (spatial adaptivity) that
neural networks can overcome, distinguishing them from some established estimators.

**Reviewer #4: Paper is poorly written/hard to follow... notation is unclear/terminology is not defined.**

We carefully reread the sections the reviewer mentions (lines 45-48, Section 2.2, Section 5), and we were unable to find
the undefined notation. We would appreciate if the reviewer could specify which aspects of the organization can be
improved, and which notation/terminology can be clarified – we would be happy to make these improvements.

**formal problem statement is vague:** The formal problem is to lower and upper bound the general minimax rate
$M\left(B_{p_d,q_d}^{\sigma_d}, B_{p_g,q_g}^{\sigma_g}\right)$ (defined in (2)), as well as on its linear counterpart (defined in (4)). We did not understand why the
reviewer found this vague and would appreciate if the reviewer could clarify this – we'll be happy to clarify these.

**lines 125-156, authors never explain how exactly their results generalize all these known results:**

Each loss listed in lines 125-156 corresponds to a particular value (or set of values) of $\sigma_d$, $p_d$, and $q_d$. Our results
(specifically, Theorems 4 and 5) "generalize" these known rates in that they hold simultaneously for many different
values of $\sigma_d$, $p_d$, and $q_d$, including but not limited to those listed on lines 125-156. We'll add this explanation to paper.

**...as if parts of the paper were taken directly from a textbook without a proper introduction of terminology.**

To our knowledge, convergence rates under all these losses have not been studied in a unified setting. Many of the
results cited here are from within the last year, and we hope this section can help consolidate this very recent work.

Due to space constraints, we omit some common alternative definitions of these losses (e.g., optimal-transport def. of
Wasserstein metric), but they are equivalently defined in terms of Besov spaces in the way we give (e.g., Wasserstein
metric is equivalent to $d_{B_{\infty,\infty}^1}$). We include common alternative notation (e.g., $C^1(1)$ on line 132) without definition to
help relate to these defs with which the reader may be familiar, but emphasize that these alternative notations are *not*
needed to understand the paper. If reviewer finds this confusing, we can remove these notations or define in appendix.

**...need to clearly demonstrate relevance of new results for GANs and nonparametric density estimation.**

While the paper's focus is theoretical, its results are relevant to both density estimation and GAN literatures:

**Relevance for density estimation:** Usually, density estimation is not performed in isolation, but rather as a sub-routine.
Hence, importance of our theory for density estimation is best seen in downstream applications. Two examples:

1. In distributionally robust optimization, a bound on the convergence rate of density estimation is directly used
to tune the optimizer (see, e.g., Esfahani & Kuhn (2015) or Staib & Jegelka (2019)).
2. Risk bounds for density estimation can be used to derive risk bounds for estimating simpler properties of
densities (such as smooth functionals; see, e.g., Kandasamy et. al (NIPS, 2015)).

Thus, our results might enable the creation of both new tools and new theoretical analyses, based on Besov IPMs.

**Relevance for GANs:** Our results are among the first finite-sample guarantees for GANs for a large family of
distributions, for which we show that well-optimized GANs are minimax optimal – results that were previously
unknown to the community. Even many simpler theoretical properties of GANs are unknown; e.g., weak consistency
was only recently studied (in [31]), without investigating convergence rates.

Besides establishing basic theoretical properties of GANs, some ways our work might contribute to GANs include:

1. Suggesting why GANs can perform density estimation well in high dimensions (by implicitly using a weak
loss, under which minimax rates might not be exponentially bad with dimension).
2. Suggesting why GANs can strictly outperform many (i.e., linear) classical density estimators.

[Meta-Review · NeurIPS 2019]

The reviewers agree that this will make a good contribution to NeurIPS. Please read the reviewer suggestions and try to incorporate them into the final submission. [This meta-review was reviewed and revised by the Program Chairs]